# ARM: Refining Multivariate Forecasting with Adaptive Temporal-Contextual Learning

**Jiecheng Lu**[1]**, Xu Han**[2]**, Shihao Yang**[1]
[1]Georgia Institute of Technology [2]Amazon Web Services
`jlu414@gatech.edu, icyxu@amazon.com, shihao.yang@isye.gatech.edu`

## Abstract

Long-term time series forecasting (LTSF) is important for various domains but is confronted by challenges in handling the complex temporal-contextual relationships. As multivariate input models underperforming some recent univariate counterparts, we posit that the issue lies in the inefficiency of existing multivariate LTSF Transformers to model series-wise relationships: the characteristic differences between series are often captured incorrectly. To address this, we introduce ARM: a multivariate temporal-contextual adaptive learning method, which is an enhanced architecture specifically designed for multivariate LTSF modelling. ARM employs Adaptive Univariate Effect Learning (AUEL), Random Dropping (RD) training strategy, and Multi-kernel Local Smoothing (MKLS), to better handle individual series temporal patterns and effectively learn inter-series dependencies. ARM demonstrates superior performance on multiple benchmarks without significantly increasing computational costs compared to vanilla Transformer, thereby advancing the state-of-the-art in LTSF. ARM is also generally applicable to other LTSF architecture beyond vanilla Transformer.

## 1 Introduction

Long-term time series forecasting (LTSF) is a critical task across various fields such as finance, epidemiology, electricity, and traffic, aiming to predict future values over an extended horizon, thereby facilitating optimal decision-making, resource allocation, and strategic planning (Martínez-Álvarez et al., 2015; Lana et al., 2018; Kim, 2003; Yang et al., 2015; Ma et al., 2022). However, modeling LTSF poses numerous challenges due to the complex and entangled characteristics of multivariate time series data, often leading to overfitting and erroneous pattern learning, thereby compromising model performance (Peng & Nagata, 2020; Cao & Tay, 2003; Sorjamaa et al., 2007).

Recently, Transformers (Vaswani et al., 2017) have significantly outperformed other structures in sequential modeling. They have showcased advancements in LTSF modeling (Zhou et al., 2022; Wu et al., 2021; Zhou et al., 2021; Li et al., 2019). While models with multivariate time series inputs are generally considered effective for capturing both temporal and contextual relationship, recent studies have shown that LTSF models with univariate input can surprisingly outperform their multivariate counterparts (Zeng et al., 2023; Nie et al., 2022). This is counter-intuitive, as univariate models are limited in capturing relationships across multiple series, which is crucial for LTSF.

In this study, we argue that existing multivariate LTSF Transformers fall short in properly modeling series-wise relationships due to suboptimal training and data processing methods, leading to subpar performance compared to univariate models. These models struggle to handle significant differences of characteristic across various input series, such as the differences in temporal dependencies, differences in local temporal patterns, and differences in series-wise dependencies beyond intra-series relationships, as exemplified in Figure 1. Improper mixing of these distinct series within crucial components of the Transformer blocks, such as temporal attention and input embedding, undermines the model's capacity to differentiate them effectively during forecasting tasks.

To address these issues, we introduce ARM: a multivariate temporal-contextual adaptive learning method. ARM incorporates improved training and processing methods to effectively manage data with pronounced series-wise characteristic differences. This leads to better handling of contextual

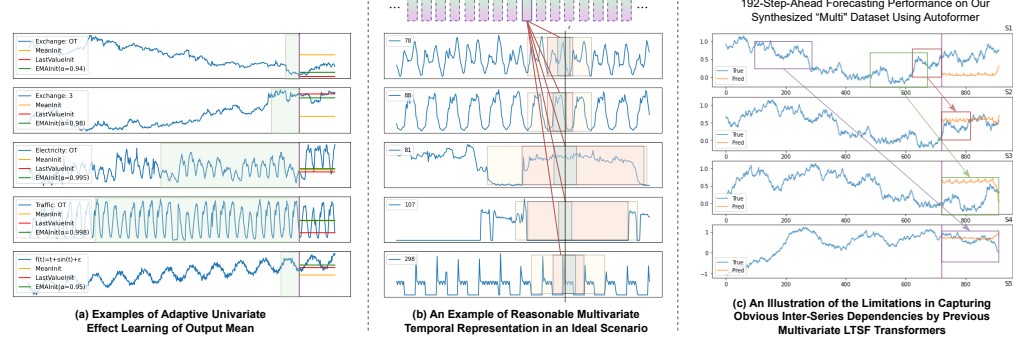

Figure 1: Three intuitive problems arising from wrongly handling input series with characteristic differences (further explained in §A.3). (a) Adaptive estimation of output mean necessity for series with diverse characteristics (see §3.1). The Adaptive EMA (green) in our method outperforms previous approaches like RevIN (yellow) and NLinear (red) by adapting to varied temporal dependencies. (b) The necessity of building reasonable multivariate temporal representation (see §3.3). Our multi-window local convolutional module addresses the inadequacy of former models in modeling different local patterns across series. (c) The Inability of Previous Models to Learn Obvious Inter-Series Dependencies: Demonstrated using the "Multi" dataset generated with simple shifting (see A.2), existing models like Autoformer fail to learn this simple "copy-paste" operations. See Figure 6 for the visualization to show our ARM's performance boost in this dataset.

information and more accurate learning of inter-series dependencies. Our approaches can be easily integrated into other LTSF models, significantly enhancing the forecasting of multivariate time series with only a modest increase in computational complexity. The key contribution of ARM comprise:

(a) We introduce Adaptive Univariate Effect Learning (AUEL), designed to independently learn the univariate effects including optimal output distribution and temporal patterns for each series before the encoder-decoder. This facilitates balanced learning of both intra- and inter-series dependencies;

(b) We implement a Random Dropping (RD) strategy to help the model identify accurate inter-series forecasting contributions and avoid overfitting from learning incorrect series-wise relationships.

(c) We propose the Multi-kernel Local Smoothing (MKLS) module, aiding Transformer blocks in adapting to multivariate input with series-wise characteristic differences. This is achieved by constructing reasonable temporal representations and enhancing locality for the Transformer blocks;

## 2 RELATED WORKS

Time series forecasting has long been a popular research topic. Traditional approaches, such as the ARIMA model (Box et al., 1974) and the Holt-Winters seasonal method (Holt, 2004), provided theoretical guarantees but were limited for complex time series data. Deep learning models (Oreshkin et al., 2020; Sen et al., 2019) introduced a new era in this field. RNNs (Hochreiter & Schmidhuber, 1997; Wen et al., 2017; Rangapuram et al., 2018; Shih et al., 2019; Salinas et al., 2020; Qin et al., 2017) allowed for the summary of past information in compact internal memory states, updated recursively with new inputs at each time step. CNNs and temporal convolutional networks (TCNs) (Lai et al., 2018; Borovykh et al., 2017; van den Oord et al., 2016) further advanced the field, capturing local temporal features effectively, though with limitations on long-term dependencies.

Recently, Transformer (Vaswani et al., 2017) succeed in sequential modeling tasks with the effective attention mechanism. Transformer derivatives like LogTrans (Li et al., 2019) introduced local convolution, reducing complexity for LTSF. Similarly, Informer (Zhou et al., 2021) and Autoformer (Wu et al., 2021) extended Transformer with efficient attention and auto-correlation mechanisms respectively, while FedFormer (Zhou et al., 2022) and PyraFormer (Liu et al., 2022) utilized improved attention to achieve lower complexity. However, a recent study (Zeng et al., 2023) argued that Transformer may fail to correctly understand multivariate time series structure. PatchTST (Nie et al., 2022) address the issue with independent input but fell short in modeling multivariate correlations. CrossFormer (Zhang & Yan, 2023) tried to address with 2D attention, but its hierarchical

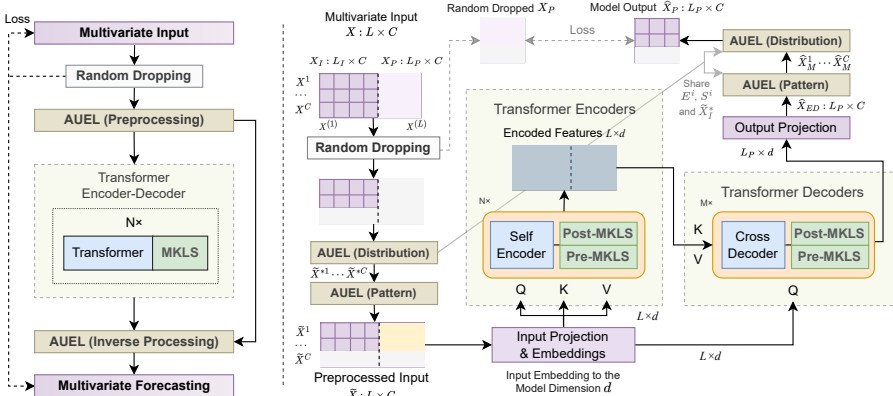

Figure 2: Overall Architecture of ARM with Vanilla Transformer as encoder-decoder. The left side depicts the global workflow and the right side illustrates the specific process in the model training.

segmentation failed adapt length per series. To address the challenges, we propose ARM to enhance the accuracy and adaptability of LTSF with proper training and data processing methods.

# 3 METHOD

We first define the notation for LTSF. For multivariate time series $X \in \mathbb{R}^{L \times C}$, our goal is to provide the best predictor $\widehat{X}_P$ of its latter part $X_P \in \mathbb{R}^{L_P \times C}$, based on its previous input part $X_I \in \mathbb{R}^{L_I \times C}$, where $L = L_I + L_P$ denotes the overall length of the time series, and $C$ represents the total number of series. Let $X^{(t),i}$ denote the value of the $t$-th step in the $i$-th sequence of $X$, where $t \in \{1, \cdots, L\}$ and $i \in \{1, \cdots, C\}$. To address the shortcomings inherent in the training of existing multivariate LTSF models—namely, their inability to effectively manage multivariate inputs with characteristic differences—we introduce the multivariate temporal-contextual adaptive method, ARM, which incorporates Adaptive Univariate Effect Learning (§3.1), Random Dropping strategy (§3.2), and Multi-kernel Local Smoothing (§3.3) on the basis of vanilla Transformer encoder-decoder structure to handle such complexities. The overall architecture of our method is illustrated in Figure 2.

## 3.1 MODULE A: ADAPTIVE UNIVARIATE EFFECT LEARNING

Determining the most likely output distribution and temporal patterns at $L_P$ based on the input at $L_I$ for different series is crucial for the accurate training of multivariate LTSF models, especially when the distribution and patterns among the series vary significantly. We refer to this operation as "Univariate Effect Learning" and introduce Adaptive Univariate Effect Learning (AUEL), specifically designed to determine the most appropriate methods to disentangle the univariate effects of different series with varying characteristics before the multivaraite encoder-decoder structure.

In previous LTSF Transformers such as Informer and Autoformer (Wu et al., 2021; Zhou et al., 2021), the $L_P$ region of the decoder input is filled with 0 values as default outputs (models without an $L_P$ part input can also be considered to use 0 default). However, the actual meaning of 0 can vary significantly across series due to their distributional differences, making it difficult for multivariate models to determine the output level for each series. As the output level predominantly influences the loss like MSE during training, the model tends to use the majority of parameters to learn the output level, making it challenging to capture finer-grained temporal patterns and inter-series dependencies.

Recent studies tried to mitigate the aforementioned issues by focusing on "eliminating distribution shifts between input and output". These approaches attempt to estimate the mean and variance, eliminating them from the model input, and restoring these values at the output. RevIN (Kim et al., 2022) calculates the mean and variance for each input series while NLinear (Zeng et al., 2023) simply employs the last value of each series at $L_I$ as the output mean. However, as illustrated in Figure 1 (a), for series with differing statistical properties, the lookback lengths required to determine their output distribution tend not to be constant. Relying solely on last or all input values will weaken per-

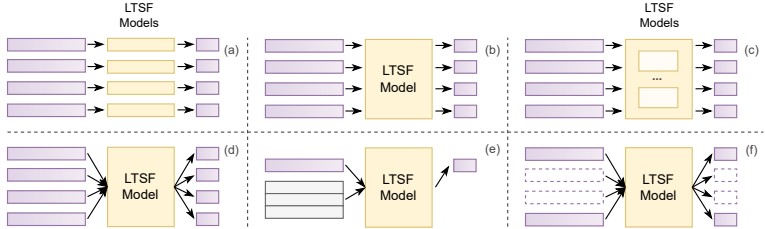

Figure 3: This figure illustrates six LTSF model types, categorized as univariate (Methods a-c) and multivariate (Methods d-f). The figure's parallel arrows show independent series processing, and converging arrows indicate series mixing. Univariate models (a-c) process series separately: (a) employs individual models like DLinear for each series, (b) uses shared-parameter models such as PatchTST for all series, and (c) dynamically chooses best models per series, as in §3.1. Multivariate models (d-f) capture inter-series dependencies: (d) uses standard structures as Informer and Autoformer, outperformed by univariate methods; (e) combines univariate models with multivariate factors to better build inter-series dependencies but adding computational complexity; (f) introduces our Random Dropping strategy (see §3.2) for efficient learning of inter-series relationships.

formance. Thus, we propose an adaptive method based on a learnable exponential moving average (EMA), which dynamically adjusts the output mean and variance for different series.

Previous methods still employ constant value to fill the $L_P$ part of each series without building temporal patterns. In multivariate LTSF, handling the univariate temporal patterns prior to the multivariate modeling can help to learn inter-series relationships more accurately. In multivariate datasets, suppose a given series $i$ has no relationships with other series. If we utilize a univariate model to initialize $X_P^i$ part based on $X_I^i$ before other operations, this preliminary step will allow the model to focus more on capturing other inter-series relationships. Previous works like LSTNet (Lai et al., 2018) and ES-RNN (Smyl, 2020) have used classic autoregressive and exponential smoothing models for this purpose. However, both methods are limited in capturing basic patterns and suffer from poor generalization. Therefore, we use an adaptive method based on the Mixture of Experts (MoE) for temporal patterns to better handle these complexities, as illustrated in Figure 3 (c).

**AUEL of Mean and Standard Deviation** AUEL uses an adaptive EMA method to learn the output mean. It can adjust the lookback length for different series using trainable EMA parameter $\alpha^i$ for each series $i$. To calculate the output standard deviation for each series, A multi-window weighting method is used. We assign $k$ windows of different lengths $w_j$ and separately compute the standard deviation of the data covered by each window. We assign trainable window weights $p_j^i$ for each series to calculate the weighted average of standard deviations across windows, resulting in $S^i$. The EMA mean $E^i$ for the $i$-th input series $X_I^i$ and its $S^i$ are computed:

$$E^i = \sum_{j=1}^{L_I} \left[ \frac{(1-\alpha^i)\exp(L_I \mathbf{1} - \mathbf{t})}{\sum_{t=1}^{L_I} \exp([L_I \mathbf{1}]_t - \mathbf{t}_t)} \circ X_I^i \right]_j, \quad S^i = \frac{\sum_{j=1}^{k} p_j^i \cdot \texttt{std}(X_I^{(L_I - w_j : L_I),i})}{\sum_{j=1}^{k} p_j^i} \tag{1}$$

where $\circ$ denotes element-wise multiplication, $\mathbf{t} = [1, 2, \cdots, L_I]$ is a vector representing the timesteps, and $[\cdot]_j$ indicates the $j$-th element of the vector enclosed in brackets; $X_I^{(L_I - w_j : L_I),i}$ is a vector composed of the part of input series $X_I^i$ that goes back $w_j$ steps from the last value. When the $\alpha^i$ of channel $i$ approaches 1, $p_{global}^i = 1$, and $p_{local}^i = 0$, the AUEL of output distribution becomes RevIN. When $\alpha^i$ approaches 0, the AUEL will be similar to NLinear which uses last values.

**AUEL of Temporal Patterns** Figure 3 parts (a) and (b) illustrate two types of univariate input LTSF models: (b) with fully-shared parameters fits datasets with similar series characteristics while (a) with independent channels is suited data with diverse series. Typically, a dataset may contain multiple clusters of similar series. Thus, (c), an intermediary approach between (a) and (b) is preferable. We use MoE, similar to the module in Switch Transformer (Fedus et al., 2022), to dynamically select the predictor based on series characteristics to initialize the temporal patterns for each series (**see A.6.1** for more details on the implementation of MoE for temporal pattern learning).

**Preprocessing and Inverse Processing** After extracting $E^i$ and $S^i$ based on $X_I^i$, we perform the AUEL of output distribution upon the input series $X^i = \begin{bmatrix} X_I^i & \mathbf{0}_p^i \end{bmatrix}$, where $\mathbf{0}_p^i$ is the 0-filled $X_P$, and $[A \ B]$ denotes the concatenation of multivariate time series $A$ and $B$ along the time dimension,

yielding the preprocessed $\widetilde{X}^{*i}$. Simillar to instance normalization (Ulyanov et al., 2016), we further apply a trainable channel affine transformation to each channel. Further, we utilize a MoE to obtain the final input $\widetilde{X}^i$. In this context, the MoE takes an input of length $L_I$ and produces an output of length $L_P$. We use $\gamma^i$, $\beta^i$ for trainable affine parameters, and $\epsilon$ for a small value to avoid overflow.

$$\widetilde{X}^{*i} = \gamma^i \left( \frac{X^i - E^i}{S^i + \epsilon} \right) + \beta^i, \quad \widetilde{X}^i = \left[ \widetilde{X}_I^{*i} \quad \text{MoE}\left( \left[ \widetilde{X}_I^{*i} \quad \widetilde{X}_P^{*i} \right] \right) \right] \tag{2}$$

We denote the forecasting for series $i$ from the encoder-decoder as $\widehat{X}_{ED}^i$. In the inverse processing stage, we again employ MoE to assist in balancing intra-series and inter-series dependencies, yielding the MoE output $\widehat{X}_M^i$. Subsequently, we reinstate $E^i$ and $S^i$ into the final output $\widehat{X}_P^i$.

$$\widehat{X}_M^i = \text{MoE}\left( \left[ \widetilde{X}_I^{*i} \quad \widehat{X}_{ED}^i \right] \right), \quad \widehat{X}_P^i = \left( \frac{(\widehat{X}_M^i - \beta^i)}{\gamma^i} \right) \cdot (S^i + \epsilon) + E^i \tag{3}$$

## 3.2 MODULE R: RANDOM DROPPING AND INTER-SERIES DEPENDENCY LEARNING

In LTSF, accurately modeling inter-series dependencies is often challenging due to the simultaneous handling of complex temporal and contextual information. Figure 1 (c) provides an illustrative example where a multivariate Transformer fails to capture some very evident inter-series dependencies.

Multivariate LTSF models often struggle to achieve disentanglement between series. Transformers with both multivariate inputs and outputs, as shown in Figure 3 (d), are prone to fitting wrong series-wise relationships. In recent research, such architectures were outperformed on datasets with commonly weaker inter-series dependencies by the two univariate model structures depicted in Figure 3 (a) and (b). To mitigate overfitting, some models combine the advantages of univariate structures, employing the architecture shown in Figure 3 (e). These models make predictions for each series individually, treating other series as context to aid the forecasting. However, when the number of series is large, the approach in (c) significantly increases computational complexity compared to multivariate LTSF models in (d), especially for the complex Transformer. It is also hard for existing structures to discern which series in the context contribute most to the forecasting of current series.

We introduce a Random Dropping to train multivariate LTSF models effectively. This strategy facilitates the learning of inter-series dependencies by efficiently decoupling relationships between different series, without increasing computational complexity. Applied during the training phase, Random Dropping simultaneously sets a random subset of series to zero in both the input and training target, as illustrated in Figure 3 (f). The model thus learns the contributions of the current subset to the forecasting of series within that subset. Through this random selection, the model incrementally discerns which series significantly influence the forecasting of others, thereby effectively mitigating the risk of overfitting. Conceptually, Random Dropping can be viewed as a form of model ensemble that constructs a pool of forecasting models for every possible subset-to-subset relationship, then enabling the more effective models among them. Utilizing Random Dropping can straightforwardly enhance the learning ability of multivariate LTSF models in terms of inter-series dependency, achieving disentanglement between series.

## 3.3 MODULE M: MULTI-KERNEL LOCAL SMOOTHING

In NLP tasks with Transformers, each input vector represents an individual word on each time step. Attention is used to compute the temporal relationships among these words. However, in the context of multivariate LTSF, the temporal dependencies for forecasting different series are typically not identical. Directly calculating attention for each input $X^{(t)}$ along the temporal dimension may lead to incorrect, blended result. This necessitates method to construct reasonable temporal representations for multivariate input. Moreover, attention assumes equal relevance between different time steps, overlooking the significance of local patterns on different series. These issues are exemplified in Figure 1 (b), which discusses the reasonable representation of multiple series. Here, building multivariate temporal representation may require different local views adjusted for different series.

In order to enhance the understanding of multivariate temporal structure, We propse the Multi-kernel Local Smoothing (MKLS) block which is used in conjunction with the Transformer blocks, as shown in Figure 4. MKLS uses multiple different 1D convolutional kernels and a channel-wise attention to learn and extract local information with adjustable view length. To address the aforementioned

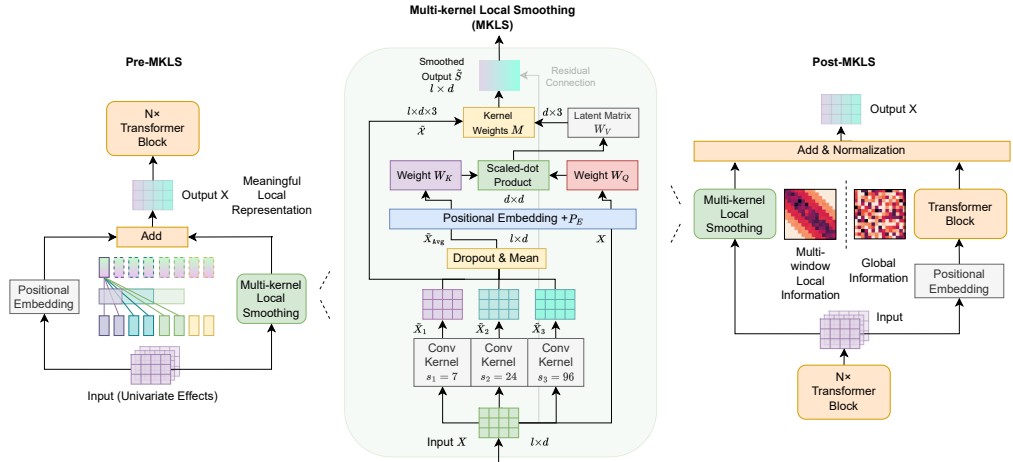

Figure 4: Structure of Multi-kernel Local Smoothing (MKLS). The central part of the figure illustrates the computation of MKLS, incorporating multiple 1D convolutions and channel attention. The left side and right side presents the application method of Pre-MKLS and Post-MKLS, respectively.

challenges of building reasonable temporal representations and enhancing locality, we provide two usage of MKLS. (i) Pre-MKLS is applied before feeding data into the Transformer blocks, Ahelping build meaningful local representations for multivariate input. (ii) Post-MKLS processes data in parallel with the Transformer blocks, playing a role like local attention with adjustable local windows. Through the incorporation of channel-wise attention, MKLS learns different kernel weights, providing robust adaptability for series with strong characteristic differences.

**MKLS block**    Let the input for the MKLS block be $X \in \mathbb{R}^{l \times d}$, where $l$ is the length of the temporal dimension and $d$ is the number of channels. Suppose we set $n$ 1D convolutional layers with kernel sizes $s_j \in (s_1, \cdots, s_n)$. First, we obtain the computation results $\widetilde{X}_j$ from the $j$-th convolutional layer by: $\widetilde{X}_j = \texttt{Conv1d}_j(X)$. To acquire the weights $m_j^i$ for kernel $j$ on channel $i$, we first perform kernel dropout and average-pooling on $\widetilde{X}_j$, and then calculate the cross attention between the averaged output and the original $X$. For kernel dropout, we randomly drop a certain proportion $r_k$ of kernel outputs, which enable independent learning of local patterns thus prevent overfitting. Let the tensor $\widetilde{\mathcal{X}} \in \mathbb{R}^{l \times d \times n}$ contain the results of each $\widetilde{X}_k \in \mathbb{R}^{l \times d}$. We now perform average-pooling to obtain another input $\widetilde{X}_{\texttt{Avg}} = \frac{1}{1-r_k} \cdot \frac{\sum_{k=1}^{n} \widetilde{X}_k}{n}$ for the attention calculation besides original $X$. Then, we use $\widetilde{X}_{\texttt{Avg}}$ and $X$ as Q and K in the attention, and obtain the kernel weights $M \in \mathbb{R}^{d \times n}$ for each channel using a trainable latent V. The specific calculation is: $M = \texttt{softmax}\left(\frac{W_Q(X^{\top} + P_E)\left(W_K(\widetilde{X}_{\texttt{Avg}}^{\top} + P_E)\right)^{\top}}{\sqrt{d}}\right) W_V$ , where $W_Q \in \mathbb{R}^{d \times d}$ and $W_K \in \mathbb{R}^{d \times d}$ are the weights for the Q and K in the attention, initialized as identity matrices to retain the original channel structure before training; $W_V \in \mathbb{R}^{d \times n}$ is designed to be a latent matrix, which projects the cross-channel attention map to the dimension of $n$, obtaining the kernel weights $M$; $P_E$ is the trainable positional embedding matrix. Now, we can calculate the weighted kernel output $\widetilde{S}$ as: $\widetilde{S}_{t,i} = \sum_{j=1}^{n} \left(\widetilde{\mathcal{X}}_{t,i,j} * M_{i,j}\right)_{t,i,j} + X_{t,i}$, where $*$ represents the element-wise multiplication performed on the channel and multi-kernel dimensions; the output $\widetilde{S}_{t,i}$ is the local smoothed output after weighting $n$ local kernels. A residual shortcut of the input is added and denoted by $X_{t,i}$.

**Pre-MKLS and Post-MKLS**    MKLS module, used alongside Transformer blocks, effectively manages local shape extraction and alignment in multivariate input. Pre-MKLS constructs local smoothed representations, offering more self-adjustment based on input compared to the pure 1D convolutional approach like (Li et al., 2019). Post-MKLS, processing data in parallel with the Transformer, enables the latter to focus more on learning long-term connections, with its logic similar to the local attention methods such as (Child et al., 2019; Roy et al., 2021) and parallel local convolutional layer in (Wu et al., 2020), but with adjustable local view. The using methods of Pre-MKLS and Post-MKLS are shown on the left side and right side of Figure 4, respectively.

Table 1: Multivariate results with prediction horizon $L_P \in \{96, 192, 336, 720\}$ (and $L_P \in \{24, 36, 48, 60\}$ for ILI). The best results are highlighted bold and the second best are underlined. Results for baseline models are sourced from previous literature or derived from additional experiments conducted for unreported datasets. Baselines were run with $L_I$ values of 96, 192, 336, 720, and best results among them are reported. In contrast, We use a consistent $L_I$ of 720 for ARM.

| Models | ARM (Vanilla) | | PatchTST | | DLinear | | FEDformer | | Autoformer | | Informer | | Repeat | |
|---|---|---|---|---|---|---|---|---|---|---|---|---|---|---|
| Metric | MSE | MAE | MSE | MAE | MSE | MAE | MSE | MAE | MSE | MAE | MSE | MAE | MSE | MAE |
| Electricity (96) | **0.125** | **0.222** | 0.129 | **0.222** | 0.140 | 0.237 | 0.193 | 0.308 | 0.201 | 0.317 | 0.274 | 0.368 | 1.588 | 0.946 |
| Electricity (192) | **0.142** | **0.239** | 0.147 | 0.240 | 0.153 | 0.249 | 0.201 | 0.315 | 0.222 | 0.334 | 0.296 | 0.386 | 1.595 | 0.950 |
| Electricity (336) | **0.154** | **0.251** | 0.163 | 0.259 | 0.169 | 0.267 | 0.214 | 0.329 | 0.231 | 0.338 | 0.300 | 0.394 | 1.617 | 0.961 |
| Electricity (720) | **0.179** | **0.275** | 0.197 | 0.290 | 0.203 | 0.301 | 0.246 | 0.355 | 0.254 | 0.361 | 0.373 | 0.439 | 1.647 | 0.975 |
| ETTm1 (96) | **0.287** | **0.340** | 0.293 | 0.346 | 0.299 | 0.343 | 0.326 | 0.390 | 0.510 | 0.492 | 0.626 | 0.560 | 1.214 | 0.665 |
| ETTm1 (192) | **0.328** | **0.364** | 0.333 | 0.370 | 0.335 | 0.365 | 0.365 | 0.415 | 0.514 | 0.495 | 0.725 | 0.619 | 1.261 | 0.690 |
| ETTm1 (336) | **0.364** | **0.384** | 0.369 | 0.392 | 0.369 | 0.386 | 0.392 | 0.425 | 0.510 | 0.492 | 1.005 | 0.741 | 1.283 | 0.707 |
| ETTm1 (720) | **0.411** | **0.412** | 0.416 | 0.420 | 0.425 | 0.421 | 0.446 | 0.458 | 0.527 | 0.493 | 1.133 | 0.845 | 1.319 | 0.729 |
| ETTm2 (96) | **0.163** | **0.254** | 0.166 | 0.256 | 0.167 | 0.260 | 0.203 | 0.287 | 0.255 | 0.339 | 0.365 | 0.453 | 0.266 | 0.328 |
| ETTm2 (192) | **0.218** | **0.290** | 0.223 | 0.296 | 0.224 | 0.303 | 0.269 | 0.328 | 0.281 | 0.340 | 0.533 | 0.563 | 0.340 | 0.371 |
| ETTm2 (336) | **0.265** | **0.324** | 0.274 | 0.329 | 0.281 | 0.342 | 0.325 | 0.366 | 0.339 | 0.372 | 1.363 | 0.887 | 0.412 | 0.410 |
| ETTm2 (720) | **0.357** | **0.382** | 0.362 | 0.385 | 0.397 | 0.421 | 0.421 | 0.415 | 0.422 | 0.419 | 3.379 | 1.388 | 0.521 | 0.465 |
| ETTh1 (96) | **0.366** | **0.391** | 0.370 | 0.400 | 0.375 | 0.399 | 0.376 | 0.415 | 0.435 | 0.446 | 0.941 | 0.769 | 1.295 | 0.713 |
| ETTh1 (192) | **0.402** | 0.421 | 0.413 | 0.429 | 0.405 | **0.416** | 0.423 | 0.446 | 0.456 | 0.457 | 1.007 | 0.786 | 1.325 | 0.733 |
| ETTh1 (336) | **0.421** | **0.431** | 0.422 | 0.440 | 0.439 | 0.443 | 0.444 | 0.462 | 0.486 | 0.487 | 1.038 | 0.784 | 1.323 | 0.744 |
| ETTh1 (720) | **0.437** | **0.459** | 0.447 | 0.468 | 0.472 | 0.490 | 0.469 | 0.492 | 0.515 | 0.517 | 1.144 | 0.857 | 1.339 | 0.756 |
| ETTh2 (96) | **0.264** | **0.327** | 0.274 | 0.337 | 0.289 | 0.353 | 0.332 | 0.374 | 0.332 | 0.368 | 1.549 | 0.952 | 0.432 | 0.422 |
| ETTh2 (192) | **0.327** | **0.374** | 0.341 | 0.382 | 0.383 | 0.418 | 0.407 | 0.446 | 0.426 | 0.434 | 3.792 | 1.542 | 0.534 | 0.473 |
| ETTh2 (336) | 0.356 | 0.393 | **0.329** | **0.384** | 0.448 | 0.465 | 0.400 | 0.447 | 0.477 | 0.479 | 4.215 | 1.642 | 0.591 | 0.508 |
| ETTh2 (720) | **0.371** | **0.408** | 0.379 | 0.422 | 0.605 | 0.551 | 0.412 | 0.469 | 0.453 | 0.490 | 3.656 | 1.619 | 0.588 | 0.517 |
| Weather (96) | **0.144** | **0.193** | 0.149 | 0.198 | 0.176 | 0.237 | 0.217 | 0.296 | 0.266 | 0.336 | 0.300 | 0.384 | 0.259 | 0.254 |
| Weather (192) | **0.189** | **0.240** | 0.194 | 0.241 | 0.220 | 0.282 | 0.276 | 0.336 | 0.307 | 0.367 | 0.598 | 0.544 | 0.309 | 0.292 |
| Weather (336) | **0.232** | **0.280** | 0.245 | 0.282 | 0.265 | 0.319 | 0.339 | 0.380 | 0.359 | 0.395 | 0.578 | 0.523 | 0.377 | 0.338 |
| Weather (720) | **0.296** | **0.332** | 0.314 | 0.334 | 0.323 | 0.362 | 0.403 | 0.428 | 0.419 | 0.428 | 1.059 | 0.741 | 0.465 | 0.394 |
| Traffic (96) | **0.356** | 0.247 | 0.360 | 0.249 | 0.410 | 0.282 | 0.587 | 0.366 | 0.613 | 0.388 | 0.719 | 0.391 | 2.723 | 1.079 |
| Traffic (192) | **0.373** | 0.258 | 0.379 | 0.256 | 0.423 | 0.287 | 0.604 | 0.373 | 0.616 | 0.382 | 0.696 | 0.379 | 2.756 | 1.087 |
| Traffic (336) | **0.383** | 0.274 | 0.392 | 0.264 | 0.436 | 0.296 | 0.621 | 0.383 | 0.622 | 0.337 | 0.777 | 0.420 | 2.791 | 1.095 |
| Traffic (720) | **0.425** | 0.294 | 0.432 | 0.286 | 0.466 | 0.315 | 0.626 | 0.382 | 0.660 | 0.408 | 0.864 | 0.472 | 2.811 | 1.097 |
| Exchange (96) | **0.078** | **0.197** | 0.087 | 0.207 | 0.081 | 0.203 | 0.148 | 0.278 | 0.197 | 0.323 | 0.847 | 0.752 | 0.081 | 0.196 |
| Exchange (192) | **0.150** | **0.280** | 0.194 | 0.316 | 0.157 | 0.293 | 0.271 | 0.380 | 0.300 | 0.369 | 1.204 | 0.895 | 0.167 | 0.289 |
| Exchange (336) | **0.252** | **0.367** | 0.351 | 0.432 | 0.305 | 0.414 | 0.460 | 0.500 | 0.509 | 0.524 | 1.672 | 1.036 | 0.305 | 0.396 |
| Exchange (720) | **0.486** | **0.535** | 0.867 | 0.697 | 0.643 | 0.601 | 1.195 | 0.841 | 1.447 | 0.941 | 2.478 | 1.310 | 0.823 | 0.681 |
| ILI (24) | **1.148** | **0.699** | 1.319 | 0.754 | 2.215 | 1.081 | 3.228 | 1.260 | 3.483 | 1.287 | 5.764 | 1.677 | 6.587 | 1.701 |
| ILI (36) | **1.352** | **0.783** | 1.579 | 0.870 | 1.963 | 0.963 | 2.679 | 1.080 | 3.103 | 1.148 | 4.755 | 1.467 | 7.130 | 1.884 |
| ILI (48) | **1.497** | **0.799** | 1.553 | 0.815 | 2.130 | 1.024 | 2.622 | 1.078 | 2.669 | 1.085 | 4.763 | 1.469 | 6.575 | 1.798 |
| ILI (60) | **1.378** | **0.771** | 1.470 | 0.788 | 2.368 | 1.096 | 2.857 | 1.157 | 2.770 | 1.125 | 5.264 | 1.564 | 5.893 | 1.677 |
| Multi (96) | **0.032** | **0.125** | 0.072 | 0.202 | 0.067 | 0.190 | 0.117 | 0.261 | 0.162 | 0.313 | 0.092 | 0.219 | 0.068 | 0.189 |
| Multi (192) | **0.063** | **0.173** | 0.167 | 0.294 | 0.137 | 0.269 | 0.204 | 0.330 | 0.356 | 0.459 | 0.207 | 0.338 | 0.143 | 0.273 |
| Multi (336) | **0.164** | **0.286** | 0.314 | 0.405 | 0.238 | 0.355 | 0.313 | 0.402 | 0.572 | 0.705 | 0.284 | 0.414 | 0.264 | 0.369 |
| Multi (720) | **0.450** | **0.503** | 0.700 | 0.588 | 0.476 | 0.522 | 0.580 | 0.544 | 0.705 | 0.621 | 0.921 | 0.795 | 0.617 | 0.551 |

## 4 EXPERIMENTS AND RESULTS

We have conducted extensive experiments on 9 widely-used datasets, including the ETT(Zhou et al., 2021), Traffic, Electricity, Weather, ILI, and Exchange Rate(Lai et al., 2018) datasets (details in §A.1). We also generated a dataset "Multi," in which we employed a simple shifting operation to construct significant inter-series relationships (see Figure 1 (c), Figure 6, and §A.2). Our baseline comparison included Transformer-based models such as PatchTST(Nie et al., 2022), FED-former(Zhou et al., 2022), Autoformer(Wu et al., 2021), and Informer(Zhou et al., 2021), and a linear model DLinear from (Zeng et al., 2023). Additionally, we included a last-value "Repeat" method for comparison. We report the baseline results by referring to their original papers. For models that did not report their performance on specific datasets, we conducted the experiments correspondingly. The performance was evaluated using mean squared error (MSE) and mean absolute error (MAE). The hyperparameter settings and implementation are detailed in §A.5 and §A.6.

In Table 1, we evaluate the performance of ARM across 10 datasets. Note that the ARM here employs a vanilla Transformer encoder-decoder, as illustrated in Figure 2, which is referred to as "Vanilla". The results demonstrate that ARM consistently outperforms previous models across 10 benchmarks. Its accurately modelling of multivariate relationships leads to enhanced performance on datasets with strong inter-series relationships and larger series scales, like Electricity, and Traffic. Also, extracting univariate effects allows better performance on datasets with short-term distribution changes like Exchange and ILI. Building on the vanilla Transformer and only slightly adding the computational costs (**see §A.7**), ARM offers adaptation across diverse multivariate datasets.

The three core modules of ARM can be easily integrated into existing LTSF models without changing their structure. Replacing the encoder-decoder predictor in Figure 2 with previous models to apply ARM can effectively help them handle multivariate characteristic differences (see §A.6.2 for more details on applying ARM to existing models). The second part of Table 2 shows the improvement ARM offers to both univariate (PatchTST, DLinear) and multivariate (Vanilla, Autoformer,

Table 2: The results of applying A/R/M to more LTSF models. We run all the experiments with fixed $L_I = 720$. Using percentage comparison, We demonstrated the enhancements in average performance of the original models with the addition of A/R/M. Note that for "+R", since the sole use of Random Dropping does not impact univariate models, these results are not provided.

| Models | Vanilla | | Vanilla+ARM | | Vanilla+A | | Vanilla+R | | Vanilla+M | | Vanilla+RM | |
|---|---|---|---|---|---|---|---|---|---|---|---|---|
| Metrics | MSE | MAE | MSE | MAE | MSE | MAE | MSE | MAE | MSE | MAE | MSE | MAE |
| Electricity (96) | 0.360 | 0.434 | 0.125 | 0.222 | 0.130 | 0.228 | 0.341 | 0.416 | 0.378 | 0.453 | 0.296 | 0.384 |
| Electricity (192) | 0.377 | 0.439 | 0.142 | 0.239 | 0.148 | 0.245 | 0.349 | 0.421 | 0.386 | 0.457 | 0.294 | 0.385 |
| Electricity (336) | 0.359 | 0.430 | 0.154 | 0.251 | 0.166 | 0.264 | 0.348 | 0.419 | 0.406 | 0.468 | 0.283 | 0.374 |
| Electricity (720) | 0.388 | 0.436 | 0.179 | 0.275 | 0.187 | 0.294 | 0.344 | 0.407 | 0.365 | 0.428 | 0.296 | 0.397 |
| ETTm1 (96) | 0.917 | 0.710 | 0.287 | 0.340 | 0.298 | 0.357 | 0.744 | 0.611 | 0.834 | 0.649 | 0.713 | 0.579 |
| ETTm1 (192) | 1.142 | 0.793 | 0.328 | 0.364 | 0.330 | 0.367 | 0.973 | 0.753 | 0.959 | 0.752 | 0.749 | 0.595 |
| ETTm1 (336) | 1.009 | 0.739 | 0.364 | 0.384 | 0.369 | 0.388 | 0.872 | 0.674 | 1.005 | 0.712 | 0.758 | 0.610 |
| ETTm1 (720) | 1.211 | 0.876 | 0.411 | 0.412 | 0.427 | 0.422 | 0.873 | 0.698 | 1.087 | 0.790 | 0.815 | 0.665 |
| Average | 0.720 | 0.607 | 0.249 | 0.311 | 0.257 | 0.321 | 0.606 | 0.550 | 0.678 | 0.589 | 0.526 | 0.499 |
| **Average%** | **100.0%** | **100.0%** | **34.6%** | **51.2%** | **35.7%** | **52.9%** | **84.2%** | **90.6%** | **94.2%** | **97.0%** | **73.1%** | **82.2%** |
| Models | PatchTST | | PatchTST+ARM | | PatchTST+A | | PatchTST+R | | PatchTST+M | | PatchTST+RM | |
| Metrics | MSE | MAE | MSE | MAE | MSE | MAE | MSE | MAE | MSE | MAE | MSE | MAE |
| Electricity (96) | 0.137 | 0.236 | 0.130 | 0.225 | 0.131 | 0.228 | - | - | 0.142 | 0.252 | 0.131 | 0.233 |
| Electricity (192) | 0.152 | 0.249 | 0.145 | 0.241 | 0.147 | 0.242 | - | - | 0.163 | 0.269 | 0.147 | 0.248 |
| Electricity (336) | 0.168 | 0.265 | 0.161 | 0.257 | 0.163 | 0.258 | - | - | 0.173 | 0.283 | 0.161 | 0.261 |
| Electricity (720) | 0.207 | 0.298 | 0.191 | 0.293 | 0.201 | 0.291 | - | - | 0.205 | 0.304 | 0.197 | 0.295 |
| ETTm1 (96) | 0.297 | 0.347 | 0.285 | 0.342 | 0.291 | 0.343 | - | - | 0.295 | 0.344 | 0.289 | 0.344 |
| ETTm1 (192) | 0.332 | 0.370 | 0.328 | 0.365 | 0.332 | 0.367 | - | - | 0.339 | 0.375 | 0.332 | 0.369 |
| ETTm1 (336) | 0.367 | 0.389 | 0.362 | 0.387 | 0.363 | 0.387 | - | - | 0.370 | 0.391 | 0.364 | 0.388 |
| ETTm1 (720) | 0.416 | 0.420 | 0.414 | 0.417 | 0.415 | 0.417 | - | - | 0.421 | 0.419 | 0.415 | 0.418 |
| Average | 0.260 | 0.322 | 0.252 | 0.316 | 0.255 | 0.317 | - | - | 0.264 | 0.330 | 0.255 | 0.320 |
| **Average%** | **100.0%** | **100.0%** | **96.9%** | **98.1%** | **98.1%** | **98.4%** | **-** | **-** | **101.5%** | **102.5%** | **98.1%** | **99.4%** |
| Models | DLinear | | DLinear+ARM | | DLinear+A | | DLinear+R | | DLinear+M | | DLinear+RM | |
| Metrics | MSE | MAE | MSE | MAE | MSE | MAE | MSE | MAE | MSE | MAE | MSE | MAE |
| Electricity (96) | 0.140 | 0.237 | 0.129 | 0.227 | 0.133 | 0.232 | - | - | 0.147 | 0.256 | 0.131 | 0.230 |
| Electricity (192) | 0.153 | 0.249 | 0.143 | 0.242 | 0.150 | 0.245 | - | - | 0.165 | 0.279 | 0.148 | 0.245 |
| Electricity (336) | 0.169 | 0.267 | 0.158 | 0.262 | 0.165 | 0.262 | - | - | 0.201 | 0.317 | 0.164 | 0.262 |
| Electricity (720) | 0.204 | 0.301 | 0.187 | 0.291 | 0.203 | 0.294 | - | - | 0.228 | 0.338 | 0.192 | 0.293 |
| ETTm1 (96) | 0.307 | 0.351 | 0.289 | 0.341 | 0.300 | 0.348 | - | - | 0.305 | 0.348 | 0.304 | 0.346 |
| ETTm1 (192) | 0.339 | 0.371 | 0.327 | 0.363 | 0.337 | 0.370 | - | - | 0.344 | 0.373 | 0.338 | 0.369 |
| ETTm1 (336) | 0.369 | 0.386 | 0.365 | 0.384 | 0.367 | 0.383 | - | - | 0.376 | 0.394 | 0.364 | 0.383 |
| ETTm1 (720) | 0.425 | 0.421 | 0.413 | 0.412 | 0.423 | 0.419 | - | - | 0.435 | 0.428 | 0.422 | 0.420 |
| Average | 0.263 | 0.323 | 0.251 | 0.315 | 0.260 | 0.319 | - | - | 0.275 | 0.342 | 0.258 | 0.319 |
| **Average%** | **100.0%** | **100.0%** | **95.4%** | **97.5%** | **98.9%** | **98.8%** | **-** | **-** | **104.6%** | **105.9%** | **98.1%** | **98.8%** |
| Models | Autoformer | | Autoformer+ARM | | Autoformer+A | | Autoformer+R | | Autoformer+M | | Autoformer+RM | |
| Metrics | MSE | MAE | MSE | MAE | MSE | MAE | MSE | MAE | MSE | MAE | MSE | MAE |
| Electricity (96) | 0.337 | 0.423 | 0.132 | 0.231 | 0.131 | 0.231 | 0.207 | 0.320 | 0.221 | 0.332 | 0.200 | 0.315 |
| Electricity (192) | 0.310 | 0.399 | 0.147 | 0.241 | 0.148 | 0.246 | 0.209 | 0.319 | 0.245 | 0.353 | 0.199 | 0.313 |
| Electricity (336) | 0.329 | 0.417 | 0.156 | 0.256 | 0.158 | 0.259 | 0.228 | 0.337 | 0.257 | 0.362 | 0.214 | 0.327 |
| Electricity (720) | 0.325 | 0.419 | 0.190 | 0.289 | 0.197 | 0.291 | 0.316 | 0.412 | 0.272 | 0.379 | 0.268 | 0.372 |
| ETTm1 (96) | 0.544 | 0.497 | 0.288 | 0.346 | 0.304 | 0.355 | 0.431 | 0.458 | 0.563 | 0.489 | 0.272 | 0.356 |
| ETTm1 (192) | 0.593 | 0.517 | 0.329 | 0.370 | 0.338 | 0.374 | 0.488 | 0.468 | 0.636 | 0.513 | 0.289 | 0.371 |
| ETTm1 (336) | 0.516 | 0.482 | 0.363 | 0.389 | 0.370 | 0.393 | 0.481 | 0.470 | 0.567 | 0.503 | 0.384 | 0.407 |
| ETTm1 (720) | 0.554 | 0.519 | 0.422 | 0.421 | 0.429 | 0.423 | 0.540 | 0.512 | 0.671 | 0.568 | 0.443 | 0.435 |
| Average | 0.439 | 0.459 | 0.253 | 0.318 | 0.259 | 0.322 | 0.363 | 0.412 | 0.429 | 0.437 | 0.284 | 0.362 |
| **Average%** | **100.0%** | **100.0%** | **57.6%** | **69.3%** | **59.0%** | **70.2%** | **82.7%** | **89.8%** | **97.7%** | **95.2%** | **64.7%** | **78.9%** |
| Models | Informer | | Informer+ARM | | Informer+A | | Informer+R | | Informer+M | | Informer+RM | |
| Metrics | MSE | MAE | MSE | MAE | MSE | MAE | MSE | MAE | MSE | MAE | MSE | MAE |
| Electricity (96) | 0.922 | 0.791 | 0.137 | 0.238 | 0.139 | 0.242 | 0.402 | 0.483 | 0.832 | 0.724 | 0.272 | 0.386 |
| Electricity (192) | 0.952 | 0.792 | 0.149 | 0.251 | 0.151 | 0.252 | 0.360 | 0.454 | 0.982 | 0.775 | 0.287 | 0.393 |
| Electricity (336) | 0.954 | 0.796 | 0.158 | 0.261 | 0.159 | 0.262 | 0.410 | 0.488 | 0.730 | 0.643 | 0.303 | 0.377 |
| Electricity (720) | 0.941 | 0.793 | 0.179 | 0.276 | 0.185 | 0.293 | 0.366 | 0.447 | 0.938 | 0.787 | 0.331 | 0.413 |
| ETTm1 (96) | 0.848 | 0.666 | 0.293 | 0.347 | 0.302 | 0.354 | 0.458 | 0.460 | 0.809 | 0.623 | 0.397 | 0.454 |
| ETTm1 (192) | 0.910 | 0.702 | 0.330 | 0.367 | 0.336 | 0.372 | 0.529 | 0.521 | 0.903 | 0.705 | 0.445 | 0.467 |
| ETTm1 (336) | 0.977 | 0.735 | 0.370 | 0.391 | 0.376 | 0.397 | 0.581 | 0.540 | 0.993 | 0.732 | 0.541 | 0.528 |
| ETTm1 (720) | 1.067 | 0.776 | 0.421 | 0.420 | 0.432 | 0.425 | 0.697 | 0.613 | 1.049 | 0.775 | 0.662 | 0.603 |
| Average | 0.946 | 0.756 | 0.255 | 0.319 | 0.260 | 0.325 | 0.475 | 0.501 | 0.905 | 0.721 | 0.405 | 0.453 |
| **Average%** | **100.0%** | **100.0%** | **27.0%** | **42.2%** | **27.5%** | **43.0%** | **50.2%** | **66.3%** | **95.7%** | **95.4%** | **42.8%** | **59.9%** |

Informer) models. We used 2 datasets for experiments: the large-scale Electricity dataset with pronounced inter-series relationships and regular temporal patterns, and the smaller ETTm1 dataset with less evident inter-series links and irregular patterns. With a fixed $L_I = 720$, our experiments show ARM's consistent performance boost without the need of tuning $L_I$ like previous models. For datasets with weaker inter-series dependencies, like ETTm1, models employing univariate predictors remain slightly superior. Conversely, for datasets with strong dependencies like Electricity, multivariate predictors have a slight edge. The balanced structure of the vanilla Transformer, without over-design, performs consistently well comparing to other models after integrating ARM.

## 4.1 ABLATION STUDIES

We conduct ablation studies on the three primary modules within ARM to showcase their specific mechanisms for performance enhancement. We use same setting for comparing different models and fix $L_I = 720$. Implementation details for using and transferring the modules are provided in §A.6. Additionally, we conducted further ablation studies on the details within the modules in §A.4.

**Effects of AUEL** In the third part of Table 2, we showcase the effects of adding the AUEL module to various models. Firstly, AUEL consistently enhances performance, especially for multivariate models: existing multivariate models mixed multiple input series directly without modeling individual univariate effect. In contrast, for univariate models, AUEL's improvements arise from its adaptive distribution handling. Comparing results from two multivariate models and two univariate models, on the Electricity dataset, where inter-series dependencies are more evident and se-

ries follow similar temporal patterns, AUEL-enhanced multivariate models outperform univariate ones. This is attributed to the better recognition of the multivariate relationship after extracting the univariate effect. However, for ETTm1, where series discrepancies are more substantial, making inter-series dependencies learning more challenging, univariate models remain advantageous.

*Remark 1.* AUEL offers a foundational univariate forecast for each series, allowing models to allocate primary parameters towards modeling the more intricate inter-series dependencies.

*Remark 2.* AUEL's results also clarify why past univariate models surpassed multivariate counterparts in LTSF: given that many benchmarks featured weak or complex inter-series relationships, previous multivariate models that blended varying series struggled with accurate series-wise pattern recognition. In such scenarios, opting to neglect hard-to-identify inter-series dependencies and focus solely on modeling univariate dependencies prevents model overfitting and improves performance.

**Effects of Random Dropping** In the fourth of Table 2, we demonstrate the performance gains of multivariate models from the Random Dropping strategy. Since univariate models train each series separately, they are not affected by this strategy, thus not included. Integrating Random Dropping considerably improves the performance of predictors without noticeably increasing computational costs. The improvements are more pronounced in datasets like ETT with weaker inter-series dependencies due to more overfitting reduction. As these relationships strengthen, evident in datasets like Electricity and even more so in Multi, the improvement of Random Dropping tend to be smaller.

*Remark 3.* Random Dropping aids multivariate LTSF models by building subset-to-subset forecasting ensemble models, effectively identifying the right combination of inter-series dependencies, thus reducing the risk of overfitting from misinterpreted series-wise relationships.

**Effects of MKLS** In the fifth part of Table 2, we show the effects of integrating MKLS into various models. For multivariate input models, MKLS helps learn local patterns across series, particularly evident on datasets with pronounced inter-series relationships like Electricity. For univariate models, adding MKLS is to attempt building multivariate relationship at the token representation level, which can lead to overfitting when series-wise relationships are weak, as shown in the results.

*Remark 4.* MKLS addresses challenges in handling series with diverse local patterns by building suitable multivariate representations and enhancing locality. Due to its dynamic local window adaptability and fitting capacity, pairing MKLS with Random Dropping is recommended to prevent potential pitfalls from wrongly learned local patterns.

**Effects of MKLS with Random Dropping** In the sixth part of Table 2, we highlight the benefits of jointly employing MKLS and Random Dropping. Using both significantly enhances performance compared to their individual applications. Adding Random Dropping reduces potential overfitting during multi-window convolutional layer training and channel-wise attention calculations in MKLS. This synergy allows the MKLS+predictor combination to be trained more effectively for multivariate input. Coupled with AUEL, ARM consistently tackle diverse multivariate forecasting challenges.

*Overall Remark.* Within ARM, we employ AUEL to adaptively learn the univariate effect for distinct series, followed by utilizing MKLS and Random Dropping to guide the predictor in discerning correct inter-series connections. Integrating all modules equips the model to handle varied datasets, ensuring correct fitting even when faced with pronounced differences temporal dependencies and local patterns, or subdued inter-series connections, leading to substantial performance improvements.

## 5    CONCLUSION

In this study, we introduce ARM, a methodology designed for effectively training multivariate LTSF models. Based on three primary modules: AUEL, Random Dropping, and MKLS, ARM addresses the challenge of handling time series with significant characteristic differences. It achieves this by extracting univariate effects, building reasonable multivariate representations, and discerning effective series combinations for forecasting contribution. Remarkably, with only a minor computational increase, ARM empowers a vanilla Transformer to achieve SOTA performance in LTSF tasks. Each module within ARM can be easily integrated into other LTSF models to enhance their performance, underscoring its practical utility and potential to influence future methodology research in the field.

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

# A  APPENDIX

## A.1  DATA SOURCES

We evaluate the performance of various long-term series forecasting algorithms using a diverse set of 10 datasets. Details of these datasets are listed below:

- The ETT dataset(Zhou et al., 2021)[1] is a collection of load and oil temperature data from electricity transformers, captured at 15-minute intervals between July 2016 and July 2018. The dataset comprises four sub-datasets, namely ETTm1, ETTm2, ETTh1, and ETTh2, which correspond to two different transformers(labeled with 1 and 2) and two different resolutions(15 minutes and 1 hour). Each sub-dataset includes seven oil and load features of electricity transformers.

- The Electricity dataset[2] includes hourly electricity consumption data for 321 clients from 2012 to 2014. It has been used in various studies related to energy consumption analysis and forecasting.

- The Exchange dataset(Lai et al., 2018)[3] is a collection of daily foreign exchange rates for eight countries, covering the period from 1990 to 2016.

- The Traffic dataset[4] is a collection of hourly road occupancy rates obtained from sensors located on freeways in the San Francisco Bay area, as provided by the California Department of Transportation. The dataset spans the period from 2015 to 2016 and has been used in various studies related to traffic forecasting and analysis.

- The Weather dataset[5] comprises 21 meteorological indicators, such as air temperature and humidity, recorded every 10 minutes throughout the entire year of 2020.

- The ILI dataset[6] is a collection of weekly data on the ratio of patients exhibiting influenza-like symptoms to the total number of patients, as reported by the Centers for Disease Control and Prevention of the United States. The dataset spans the period from 2002 to 2021 and has been used in various studies related to influenza surveillance and analysis.

## A.2  TO WHAT EXTENT DO WE NEED INDEPENDENCE IN LTSF: A MULTIVARIATE DATASET WITH SIGNIFICANT DEPENDENCIES

Additionally, we generate a multivariate dataset, termed as "Multi", with strong dependencies among series to illustrate scenarios where channel-independent approaches may fail in the presence of inter-series causal relationships. This dataset comprises eight series that individually lack long-term predictability, yet when considered in relation to each other, significant dependencies become apparent. As shown in Table 1, ARM (Vanilla) outperforms univariate models and previous multivariate models in modelling inter-series dependencies in the Multi dataset, with the visualization of the forecasting results shown in Figure 6.

To build the Multi dataset, we first generate 20,000 steps of random noise $\epsilon \sim N(0, 1)$ and build a random walk process $X^1$, where $X^{(t),1} = X^{(t-1),1} + \epsilon^{(t)}$, $X^{(1),1} = \epsilon^{(1)}$. We then take 18,000 steps from the interval between 2,000 and 20,000 as our first time series $X^{(t),1}$. The remaining seven

---

[1]https://github.com/zhouhaoyi/ETDataset
[2]https://archive.ics.uci.edu/ml/datasets/ElectricityLoadDiagrams20112014
[3]https://github.com/laiguokun/multivariate-time-series-data
[4]http://pems.dot.ca.gov/
[5]https://www.bgc-jena.mpg.de/wetter/
[6]https://gis.cdc.gov/grasp/fluview/fluportaldashboard.html

series are generated as follows:

$$X^{(t),2} = X^{(t-96),1} \tag{4}$$

$$X^{(t),3} = X^{(t-192),1} \tag{5}$$

$$X^{(t),4} = X^{(t-336),1} \tag{6}$$

$$X^{(t),5} = X^{(t-720),1} \tag{7}$$

$$X^{(t),6} = \frac{1}{2}X^{(t),2} + \frac{1}{2}X^{(t),3} \tag{8}$$

$$X^{(t),7} = \frac{1}{4}X^{(t),2} + \frac{1}{4}X^{(t),3} + \frac{1}{4}X^{(t),4} + \frac{1}{4}X^{(t),5} \tag{9}$$

$$X^{(t),8} = X^{(t),2} \cdot X^{(t),3} \tag{10}$$

It can be seen that when each series is considered separately, due to their sampling from the random walk process, effective long-term forecasting is unattainable. However, with knowledge of the past trajectory of $X^{(t),1}$, we can accurately infer the values of the remaining seven series through $X^{(t),1}$.

In this dataset, we can observe that there is only one actual generative sequence, making the mixing and compressing of all channels into a lower dimension feasible. However, as seen from Table 1, although channel-independent univariate models are clearly unsuitable for this dataset, neither Autoformer nor Informer outperform the univariate models. This verifies our previous assertion to some extent: previous multivariate LTSF models likely have not learned the causal relationships between different series because of its inefficiency usage of parameters. In contrast, as we can see from Figure 6, ARM effectively learns the causal relationships between series, performing a rather perfect fit for shifting and linear, non-linear combinations of the generative series. In scenarios with strong multivariate dependencies, ARM's forecasting results significantly exceed previous multivariate univariate LTSF models.

## A.3 ADDITIONAL EXPLAINATION OF THE THREE EXAMPLES IN FIGURE 1

(a) The necessity of adaptive estimation of output mean for series with different characteristics (see §3.1): We illustrate how different methods estimate the output mean for five example time series. The blue lines denote the original series, with the input and output sections separated by a purple line. Using yellow and red lines, we represent methods that employ the input mean (RevIN) and the last input value (NLinear) respectively as the output mean. These previous methods are suboptimal as they don't account for varying lookback lengths required for different series. We highlight the ideal lookback areas with green boxes. The Adaptive EMA in our Adaptive Univariate Effect Learning method, represented by a green line, effectively learns the optimal output mean, thereby handling series with diverse temporal dependencies more accurately.

(b) The necessity of building reasonable multivariate temporal representation (see §3.3), which is demonstrated by five series from the Electricity dataset. A black vertical line is used to indicate the current time step $t$. Their associated temporal relationships and local patterns vary for different series. Thus, adopting the values from $X^{(t)}$ as the representation for time step $t$, as done in previous models, wrongly blends the modeling across series with distinct characteristics. We use green, red, and yellow boxes to depict the optimal scope of local patterns that should be considered for different series at time $t$. To accommodate these distinct patterns, we introduce a multi-window local convolutional module to construct reasonable representation for multivariate inputs.

(c) The Inability of Previous Models to Learn Obvious Inter-Series Dependencies: We illustrate the ineffectiveness of existing multivariate LTSF Transformers using a dataset "Multi", where the subsequent three series are simple shifts of the first (details in section A.2). A "copy-paste" operation should suffice for forecasting, yet previous multivariate models, such as Autoformer, fall short to accomplish this task efficiently. In Figure 6, we visually demonstrate ARM's enhancement in handling these strong inter-series relationships for LTSF models.

## A.4 ADDITONAL ABLATION STUDIES

In Table 3, we conducted ablation studies on finer details within ARM. The last row of the table presents the optimal results for ARM (Vanilla), with other rows detailing variations on this vanilla ARM to observe result changes. Some more granular analyses of ARM's effects are also provided.

### A.4.1 SUB-MODULES OF AUEL

The first section of Table 3 displays the impact of modifications in the AUEL module. Initially, we attempted to omit modules for learning distribution and temporal patterns separately; both deletions led to performance dips, especially for the latter. Such declines were more pronounced on datasets with weaker inter-series dependencies, like ETTm1. Subsequently, we replaced the MoE module in learning temporal patterns (see Figure 3 (c)) with independent linears (see Figure 3 (a)) and a single linear layer (see Figure 3 (b)). As analyzed in the main text, these substitutions diminished the effectiveness of temporal pattern learning.

### A.4.2 SUB-MODULES OF MKLS

The second section of Table 3 presents ablation studies on MKLS sub-modules. When either Pre-MKLS or Post-MKLS was removed, there was a noticeable performance degradation. Trying to eliminate the channel-wise attention in MKLS, where outputs from different convolution kernels are averaged with equal weights for each series, also affected performance.

We further examined optimal kernel size $s$ choices within MKLS. It's worth noting that for other experiments, we utilized $s = [49 \ 145 \ 385]$. Results indicate that, for $L_I = 720$, this size selection outperforms other combinations. Especially in datasets like Electricity, with numerous series and longer temporal dependencies, proper use of larger convolutional kernels somewhat improved performance.

### A.4.3 OTHER COMBINATIONS OF ARM

Considering eight possible combinations of A/R/M modules, six were addressed in the main text. In the third section of Table 3, we evaluate the remaining two combinations. Given that AUEL offers a baseline univariate solution for forecasting, combinations including the A module yield results close to the SOTA. From AM and AR combinations, it's evident that combinations solely containing M are more susceptible to overfitting. Integrating all ARM modules resulted in a performance boost compared to just using AM or AR.

Table 3: Additional ablation studies of the submodules included in AUEL and MKLS. Other two combinations of ARM modules that are not included in the result table of the main text is also presented. The experiments are conducted on the basis of the Vanilla ARM setting.

| Datasets (Predict $L_P$) | | Electricity ($L_P = 96$) | | Electricity ($L_P = 336$) | | ETTm1 ($L_P = 96$) | | ETTm1 ($L_P = 336$) | |
|---|---|---|---|---|---|---|---|---|---|
| | Metric | MSE | MAE | MSE | MAE | MSE | MAE | MSE | MAE |
| AUEL | w/o learning distribution | 0.132 | 0.232 | 0.159 | 0.262 | 0.293 | 0.349 | 0.370 | 0.391 |
| AUEL | w/o learning temporal patterns | 0.142 | 0.235 | 0.185 | 0.287 | 0.379 | 0.525 | 0.403 | 0.439 |
| AUEL | independent linears | 0.133 | 0.231 | 0.165 | 0.264 | 0.301 | 0.346 | 0.379 | 0.393 |
| AUEL | single linear | 0.128 | 0.226 | 0.159 | 0.261 | 0.307 | 0.353 | 0.384 | 0.399 |
| MKLS | w/o Pre-MKLS | 0.127 | 0.225 | 0.161 | 0.262 | 0.292 | 0.348 | 0.369 | 0.391 |
| MKLS | w/o Post-MKLS | 0.130 | 0.229 | 0.163 | 0.261 | 0.295 | 0.349 | 0.370 | 0.390 |
| MKLS | w/o attention | 0.129 | 0.227 | 0.158 | 0.255 | 0.294 | 0.347 | 0.367 | 0.389 |
| MKLS | kernel $s = [49]$ | 0.129 | 0.226 | 0.163 | 0.260 | 0.287 | 0.347 | 0.367 | 0.389 |
| MKLS | kernel $s = [49 \ 145]$ | 0.129 | 0.230 | 0.164 | 0.262 | 0.292 | 0.345 | 0.366 | 0.387 |
| MKLS | kernel $s = [49 \ 145 \ 385 \ 673]$ | 0.127 | 0.224 | 0.156 | 0.255 | 0.297 | 0.349 | 0.365 | 0.389 |
| A/R/M | Vanilla+AR | 0.132 | 0.232 | 0.165 | 0.263 | 0.298 | 0.349 | 0.372 | 0.393 |
| A/R/M | Vanilla+AM | 0.136 | 0.238 | 0.167 | 0.266 | 0.321 | 0.368 | 0.386 | 0.406 |
| **Full** | **ARM (Vanilla)** | **0.125** | **0.222** | **0.154** | **0.251** | **0.287** | **0.340** | **0.364** | **0.384** |

Table 4: Evaluation metrics of previous LTSF models with only the adaptive distribution learning module in AUEL (denoted as A(D)). Forecasting input lengths $L_I$ are set to 720 for all the experiments.

| Models | | Autoformer | | Autoformer+A(D) | | DLinear | | DLinear+A(D) | | PatchTST | | PatchTST+A(D)* | |
|---|---|---|---|---|---|---|---|---|---|---|---|---|---|
| Metric | | MSE | MAE | MSE | MAE | MSE | MAE | MSE | MAE | MSE | MAE | MSE | MAE |
| Exchange | 96 | 1.139 | 0.832 | 1.012 | 0.766 | 0.141 | 0.280 | 0.104 | 0.229 | 0.087 | 0.210 | 0.081 | 0.197 |
| | 192 | 1.203 | 0.850 | 1.082 | 0.798 | 0.264 | 0.393 | 0.240 | 0.346 | 0.182 | 0.306 | 0.170 | 0.291 |
| | 336 | 1.474 | 0.932 | 1.153 | 0.818 | 2.076 | 1.095 | 0.445 | 0.478 | 0.372 | 0.446 | 0.315 | 0.402 |
| | 720 | 3.309 | 1.446 | 2.036 | 1.087 | 6.486 | 1.814 | 1.252 | 0.859 | 1.082 | 0.791 | 0.827 | 0.683 |
| Weather | 96 | 0.429 | 0.432 | 0.369 | 0.362 | 0.144 | 0.201 | 0.142 | 0.190 | 0.145 | 0.197 | 0.141 | 0.195 |
| | 192 | 0.403 | 0.420 | 0.365 | 0.363 | 0.186 | 0.251 | 0.183 | 0.231 | 0.190 | 0.239 | 0.186 | 0.239 |
| | 336 | 0.457 | 0.461 | 0.363 | 0.366 | 0.237 | 0.295 | 0.233 | 0.272 | 0.246 | 0.287 | 0.238 | 0.280 |
| | 720 | 0.746 | 0.593 | 0.378 | 0.379 | 0.306 | 0.350 | 0.305 | 0.325 | 0.309 | 0.331 | 0.308 | 0.328 |

* PatchTST previously utilized RevIN, which is replaced with the adaptive distribution learning in AUEL.

Table 5: The enhancement of the efficiency of parameter utilization provided by learning temporal patterns (MoE) in AUEL. We conducted experiments using the Vanilla model settings. We specifically evaluated the performance when the model dimension $d$ of Transformer blocks is set to 16, 64, and 256 respectively. The left three experiments employed the MoE from AUEL, while the right three did not. The overall best results are highlighted in bold, and the best results within each group are underlined.

| Model | | MoE ($d = 16$) | | MoE ($d = 64$) | | MoE ($d = 256$) | | w/o MoE ($d = 16$) | | w/o MoE ($d = 64$) | | w/o MoE ($d = 256$) | |
|---|---|---|---|---|---|---|---|---|---|---|---|---|---|
| Metric | | MSE | MAE | MSE | MAE | MSE | MAE | MSE | MAE | MSE | MAE | MSE | MAE |
| Electricity | 96 | 0.128 | 0.228 | **0.126** | **0.225** | 0.129 | 0.229 | 0.313 | 0.392 | 0.261 | 0.348 | 0.256 | 0.341 |
| Electricity | 192 | 0.146 | 0.242 | **0.142** | **0.239** | 0.145 | 0.245 | 0.299 | 0.378 | 0.255 | 0.349 | 0.251 | 0.352 |
| ETTm1 | 96 | **0.287** | **0.340** | 0.295 | 0.348 | 0.300 | 0.352 | 0.488 | 0.460 | 0.467 | 0.446 | 0.503 | 0.468 |
| ETTm1 | 192 | **0.325** | **0.371** | 0.342 | 0.376 | 0.346 | 0.385 | 0.531 | 0.499 | 0.498 | 0.482 | 0.574 | 0.512 |

### A.4.4 EFFECT OF AUEL DISTRIBUTION LEARNING

This section displays the impact of solely employing adaptive distribution learning from AUEL on existing LTSF models, as illustrated in Table 4. Adaptive learning for level and variance proves effective in scenarios with distinct series temporal dependencies and varying local distributions, as seen in the Exchange and Weather datasets. Results affirm the significant role of adaptive distribution learning in enhancing performance across irregular datasets.

### A.4.5 REDUCTION AND ROBUSTNESS IMPROVEMENT OF PARAMETER SIZE WITH ARM

In this section, we demonstrate the efficacy of AUEL in enhancing model parameter efficiency and reducing the optimal parameter size required in the main predictor, as shown in Table 5. Owing to AUEL's ability to disentangle univariate effects, the majority of parameters in the main predictor are primarily used for modeling inter-series dependencies, resulting in a significant reduction in the number of parameters needed. From the results, it is evident that on both the larger-scale Electricity dataset and the smaller ETTm1 dataset, the introduction of MoE significantly reduces the optimal model dimension $d$.

Also, with the existence of ARM, the model become insensitive to the choice of $d$: after the extraction of univariate effect by AUEL, the model parameters of other parts, whose size mainly controlled by model dimension $d$, will focus on modelling the inter-series dependencies. $d$ typically needs to be tuned to accommodate different inter-series relationship strength of datasets. A smaller $d$ helps avoid overfitting in datasets with weak inter-series relationships, whereas a larger $d$ is necessary for datasets with stronger inter-series dependencies. Incorporating Random Dropping mitigates the need for tuning $d$ across various datasets. Random Dropping can be regarded as an adaptive scaler for $d$, allowing the model to automatically adjust to the strength of these relationships. Consequently, $d$ can be set solely based on the number of input series $C$ (we use $d = 16$ for $C < 20$ and $d = 64$ for $C \geq 20$ for all the experiments), with Random Dropping modulating the learning of inter-series relationships. On datasets Exchange (weak inter-series dependencies) and Multi (strong inter-series dependencies) with both of their $C = 8$, the results in Table 6 demonstrate the effectiveness of "R": using a consistent $d = 16$ that potentially fits stronger inter-series dependencies (like Multi), models with Random Dropping (+ARM) perform equally well on both types of datasets, without the overfitting on Exchange dataset observed in models without Random Dropping (+AM).

Table 6: The insensitivity of the performance regarding the model dimension $d$ with the existence of Random Dropping. We conducted the experiments with a consistent $d = 16$ on two datasets with the same number of series $C = 8$ but different strength of inter-series relationship. Exchange dataset have weak inter-series link which inherently requires less parameters $d < 16$ to model it, while Multi dataset have stronger inter-series dependencies and fits the scenario of $d = 16$. With ARM comparing to AM, models perform equally well on datasets with both strong and weak inter-series dependencies. In other words, random dropping can provide more improvement on the datasets with weaker inter-series dependencies when $d$ is high enough based on $C$.

| Models | Autoformer+AM | | Autoformer+ARM | | Informer+AM | | Informer+ARM | | Vanilla+AM | | Vanilla+ARM | |
|---|---|---|---|---|---|---|---|---|---|---|---|---|
| Metric | MSE | MAE | MSE | MAE | MSE | MAE | MSE | MAE | MSE | MAE | MSE | MAE |
| Exchange (96) | 0.089 | 0.21 | 0.081 | 0.201 | 0.090 | 0.212 | 0.086 | 0.207 | 0.085 | 0.206 | 0.078 | 0.197 |
| Exchange (336) | 0.332 | 0.417 | 0.298 | 0.395 | 0.338 | 0.420 | 0.312 | 0.404 | 0.304 | 0.398 | 0.252 | 0.367 |
| Average% | 100.0% | 100.0% | 90.0% | 95.1% | 100.0% | 100.0% | 93.0% | 96.7% | 100.0% | 100.0% | 84.8% | 93.4% |
| Multi (96) | 0.037 | 0.130 | 0.038 | 0.130 | 0.051 | 0.164 | 0.051 | 0.166 | 0.034 | 0.129 | 0.032 | 0.125 |
| Multi (336) | 0.181 | 0.302 | 0.175 | 0.300 | 0.182 | 0.305 | 0.183 | 0.304 | 0.172 | 0.291 | 0.164 | 0.286 |
| Average% | 100.0% | 100.0% | 97.7% | 99.5% | 100.0% | 100.0% | 100.4% | 100.2% | 100.0% | 100.0% | 95.1% | 97.9% |

## A.5 MODEL TRANING AND HYPER-PARAMETERS

The ARM (Vanilla) model is trained using the Adam optimizer and MSE loss in Pytorch, with a learning rate of 0.00005 over 100 epochs on each dataset with a early-stopping patience being 30 steps. The first 10% of epochs are for warm-up, followed by a linear decay of learning rate. Owing to ARM's adaptability to long sequences, unlike baseline models, we only use a lookback window of length 720 for training (and 104 for the ILI dataset). The multi-kernel size $s$ of MKLS is set to $[25 \ 145 \ 385]$, which is also used as the multi-window size in the AUEL's adaptive learning of standard deviation.

We run our model on a single Nvidia RTX 3090 GPU. We use a batch size of 32 for most datasets. For some datasets with a larger number of series, if a longer $L_P$ renders the running unfeasible, we try to reduce the batch size to 16 or 8, respectively. For the dimension of the main part of the model, we need to adjust it accordingly based on the strength of series-wise relationship. We can mainly consider the scale of a dataset if we are not familiar with the dataset. We employ a Transformer model dimension $d = 16$ for most of the small datasets ($C < 20$) like ETTs, Exchange, ILI, and Multi with less series-wise relationship needed to be learned by our model. For the datasets like Weather, Electricity, and Traffic ($C \geq 20$), which have more series and potentially more series-wise relationship to learn, we raise the dimension $d$ to 64 (you can also refer to Table 5 for more intuition of the setting of $d$). You can basically use these two dimension settings for most of the datasets. As for the number of heads in multi-head attention, we set it to 8. For the number of layers in the Transformer encoder-decoder structure, we use two encoder layers and one decoder layer. We do not apply dropout in the Transformer Encoder; in the MKLS, we set the dropout rate to 0.25; and in the MoE, we set the dropout rate to 0.75. We use a high dropout rate for MoE to avoid overfitting because we set a relatively large $4(L_I + L_P)$ as the hidden dimension of the MoE predictors. As for the number of experts in the MoE module, we use 2 experts for the small datasets stated above and 4 experts for the larger datasets. We set the random seed for the main experiments as 2024.

For model selection, we partition each dataset into training, validation, and test sets with proportions of 70%, 10%, and 20%, respectively. The models are trained on the training set, and the best model is selected based on its MSE on the validation set. The MSE and MAE of this model on the test set are reported.

## A.6 ADDITIONAL IMPLEMENTATION DETAILS OF MODULES

### A.6.1 ADDITIONAL IMPLEMENTATION DETAILS OF AUEL

In the adaptive learning of distribution, we employ $\alpha^i = 0.9$ as the initialized EMA alpha parameter for each series. For the multi-window of adaptive standard deviation, we initialize with equal weights. Clipping operations are used to prevent the EMA alpha and multi-window weights from becoming negative.

For adaptive learning of temporal patterns, we utilize the MoE predictor structure as described in (Fedus et al., 2022). We adjusted the output dimension of its final layer to cater to the in-

put length of $L_I + L_P$, an output length of $L_P$, with a hidden dimension of $4(L_I + L_P)$. Given the discrepancy in input and output lengths, the MoE's residual shortcut needs specific modifications. We establish a residual connection using the $L_P$ section of the input before the AUEL preprocessing and before the AUEL inverse processing. Specifically, in the calculation of MoE in AUEL preprocessing, where $\widetilde{X}^i = \left[\widetilde{X}_I^{*i} \ \texttt{MoE}\left(\left[\widetilde{X}_I^{*i} \ \widetilde{X}_P^{*i}\right]\right)\right]$, we actually compute $\texttt{MoE}\left(\left[\widetilde{X}_I^{*i} \ \widetilde{X}_P^{*i}\right]\right) = \texttt{MLP}_{\texttt{selected}}\left(\left[\widetilde{X}_I^{*i} \ \widetilde{X}_P^{*i}\right]\right) + \mathbf{0}_P^i$. Here, $\texttt{MLP}_{\texttt{selected}}$ denotes the MLP predictor selected by the routing part of MoE based on the input, and $\mathbf{0}_P^i$ is the $L_P$ part before AUEL preprocessing, i.e., the default 0 values before AUEL. During inverse processing, we compute $\widehat{X}_M^i = \texttt{MoE}\left(\left[\widetilde{X}_I^{*i} \ \widehat{X}_{ED}^i\right]\right) = \texttt{MLP}_{\texttt{selected}}\left(\left[\widetilde{X}_I^{*i} \ \widehat{X}_{ED}^i\right]\right) + \widehat{X}_{ED}^i$, where the added $\widehat{X}_{ED}^i$ is the $L_P$ part before the AUEL block.

### A.6.2 Additional Implementation Details of MKLS and the Transfering of MKLS to Univariate Models

When migrating ARM to other LTSF models, both AUEL and Random Dropping can be directly applied at the input and output ends. However, as MKLS requires integration with Transformer blocks, its application becomes challenging if the main predictor lacks a Transformer structure. Hence, for MKLS transfer, we employ a approach that remains unaffected by the architecture of the main predictor. We treat the main predictor as a whole: after data undergoes AUEL preprocessing and before entering the main predictor, we use Pre-MKLS. Once we obtain the result from the main predictor and before using AUEL inverse processing, we utilize Post-MKLS. This approach, which treats the predictor holistically as a Transformer block, effectively enhances its ability to handle multivariate inputs and amplifies its locality learning capability.

In ARM (Vanilla), MKLS is applied to a latent model dimension with $d$ channels. For multivariate LTSF Transformers like Autoformer and Informer, they project input data to this latent model dimension, making their MKLS application consistent with Vanilla. However, for univariate models like DLinear and PatchTST, $X^{(t)}$ is not projected onto the latent dimension, but maintaining the original $C$ channels of input series when processing the input series. This inconsistency might skew results in model comparisons, impeding the full potential of MKLS in univariate models. Consequently, for these univariate models, we devised an mixed input and output processing method. After AUEL preprocessing, we project the $C$ series to a dimension of $d - C$ and concatenate it with these series to get a Pre-MKLS input of dimension $d$. Thus, within this $d$-dimensional input, channels from the original series ($C$) coexist with channels representing a mix of multiple series ($d - C$). After receiving the predictor's output, it is fed into Post-MKLS, then projected back to dimension $C$, merged with the previous result, and subjected to AUEL inverse processing. This strategy seamlessly integrates MKLS's capacity for handling inter-series relationships into these univariate models, while preserving the univariate models' proficiency in processing intra-series information independently, providing a fair comparison between the univariate methods and multivariate methods when implementing MKLS module on them.

### A.6.3 Additional Implementation Details of Random Dropping

Random Dropping can be regarded as a channel dropout module acting simultaneously on both the input and training target, with the dropout rate adjusted randomly at each training step. In every training iteration, we randomly generate a dropping rate $r_d$ and fill $r_d C$ of the $C$ input series with zeros. Through this method, we can explore all possible series combinations during training. By identifying groups or clusters of series with useful inter-series causal relationships in them, their forecasting contribution is gradually reinforced within the model parameters via gradient updates.

### A.6.4 Other Implementation Details

We use some types of widely-used embedding matrices adopted from in previous LTSF research. Firstly, a trainable position embedding matrix is added after performing the input projection. We also add two trainable task embedding matrices for both the $L_I$ part and $L_P$ of the input. Timestep / Date Embedding is also optional to use in our model architecture. Since these embedding matrices have very limited effects on our model results, which is similar to the observation in previous research

Table 7: Comparative Analysis of Computational Costs in LTSF Models. This comparison utilizes the ETTm1 data format for constructing model inputs.

| | Vanilla+ARM (FLOPs) | Vanilla+ARM (Params) | Vanilla (FLOPs) | Vanilla (Params) | Autoformer (FLOPs) | Autoformer (Params) |
|---|---|---|---|---|---|---|
| $X_P = 96$ | 426M | 7.89M | 244M | 14.9M | 10.9G | 15.5M |
| $X_P = 192$ | 515M | 10.4M | 273M | 17.5M | 11.6G | 15.5M |
| $X_P = 336$ | 664M | 14.9M | 316M | 21.9M | 12.7G | 15.5M |
| $X_P = 720$ | 1.15G | 30.6M | 431M | 37.8M | 15.5G | 15.5M |
| | Informer (FLOPs) | Informer (Params) | PatchTST (FLOPs) | PatchTST (Params) | DLinear (FLOPs) | DLinear (Params) |
| $X_P = 96$ | 9.41G | 11.3M | 5.26G | 4.87M | 3.06M | 138K |
| $X_P = 192$ | 10.1G | 11.3M | 5.31G | 7.08M | 6.10M | 277K |
| $X_P = 336$ | 11.2G | 11.3M | 5.38G | 10.4M | 10.7M | 485K |
| $X_P = 720$ | 14.0G | 11.3M | 5.58G | 19.2M | 22.8M | 1.04M |

like DLinear and PatchTST, we just simply enable these embeddings in our architecture without particularly emphasizing them.

## A.7 COMPUTATIONAL COSTS

We present the comparison of computational costs between our Vanilla+ARM method and the existing LTSF SOTAs, as shown in Table 7. The results are calculated using the "ptflops" package. We conduct the experiments with the same input format as the data input in ETTm1 dataset. For existing models, we build them with the best hyper-parameter settings stated in their original papers. We set the token dimension of Vanilla to 64 and $d$ of Vanilla+ARM to 16 in order to emphasize the efficiently using of predictor parameters provided by our ARM architecture: the optimal dimension required by the encoder-decoder is reduced after applying ARM, as presented in Table 5.

## A.8 PREDICTABILITY ANALYSIS

To illustrate the variability in predictability across datasets, we conduct a simple analysis in this section, as shown in Figure 5. In each of the eight datasets shown in the figure, we randomly select subseries of length 432, using the first 336 steps as the training set and the final 96 steps as the testing set to fit traditional univariate time series models. The models we employ include naive repeat, ARMA(p, q), Simple Moving Average with different lookback windows (SMA), Exponential Smoothing with different alpha values (ES), Random Forest with different lookback windows (RFR), and MLP with various hidden layer dimensions and lookback windows. We calculate the RMSE on the testing set as the performance metric for each model. To compare the relative performance of different models on various series, we normalize and invert the RMSE for each series, then standardize across the model dimension. This process ensures that a higher value corresponds to better relative model performance. We visualize the transformed performance evaluations of different models on various datasets using boxplots.

As shown in the figure, in datasets where time series with regular patterns dominate, such as Electricity, more complex models like MLP and Random Forest demonstrate superior performance for these regularities, and longer lookback lengths yield better results. In contrast, in datasets with weaker regularity and sudden changes, like Exchange and ILI, methods that either focus on a shorter lookback view or assign greater weight to recent datapoints, such as SMA and ES, perform better. This predictability analysis indicates the necessity of adaptively adjusting the lookback view and complexity of a model for time series with different characteristics. In ARM, we use AUEL, MKLS, and Random Dropping to accommodate these characteristics changes, effectively handling the significant differences observed across various datasets.

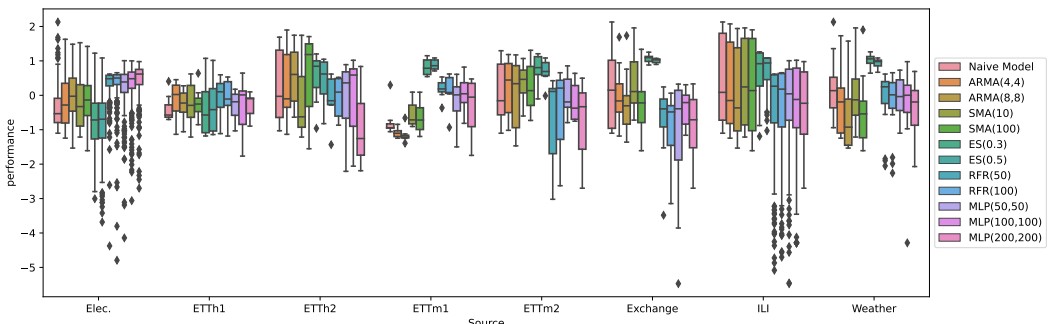

Figure 5: Predictability Analysis of Datasets (Elec. means Electricity)

## A.9 VISUALIZATION ANALYSIS

### A.9.1 VISUALIZATION OF THE FORECASTING FOR MULTI DATASET

In figure 6, we show the visualization of the best forecasting results for Vanilla, PatchTST, DLinear, and Autoformer models with and without ARM on the Multi dataset, with a prediction length (LP) of 96 steps. ARM effectively equips the LTSF models with enhanced ability of modelling inter-series shape connection.

### A.9.2 RANDOMNESS OF TRAINING

Figure 7 illustrates the impact of adjusting the random seed on the model performance. ARM integrates random training techniques, which might make the model training sensitive to the setting of random seed in small datasets. Thus, we conduct experiments to use different random seeds for the training on the small datasets with irregular patterns like Exchange, ETTh1, ETTh2, Weather. Figure 7 demonstrates that in most cases, using a fixed random seed for the training of ARM is enough to surpass previous best-performing models.

## A.10 FUTURE WORKS

For future research, potential applications of ARM modules in other time series tasks can be explored. Given that MKLS and Random Dropping decouple relationships between series, it would be worth investigating their viability in time series classification and anomaly detection. Additionally, as AUEL effectively extracts a series' intrinsic effects, enhancing the extraction of inter-series relationships, its application in tasks like time series clustering can be further explored.

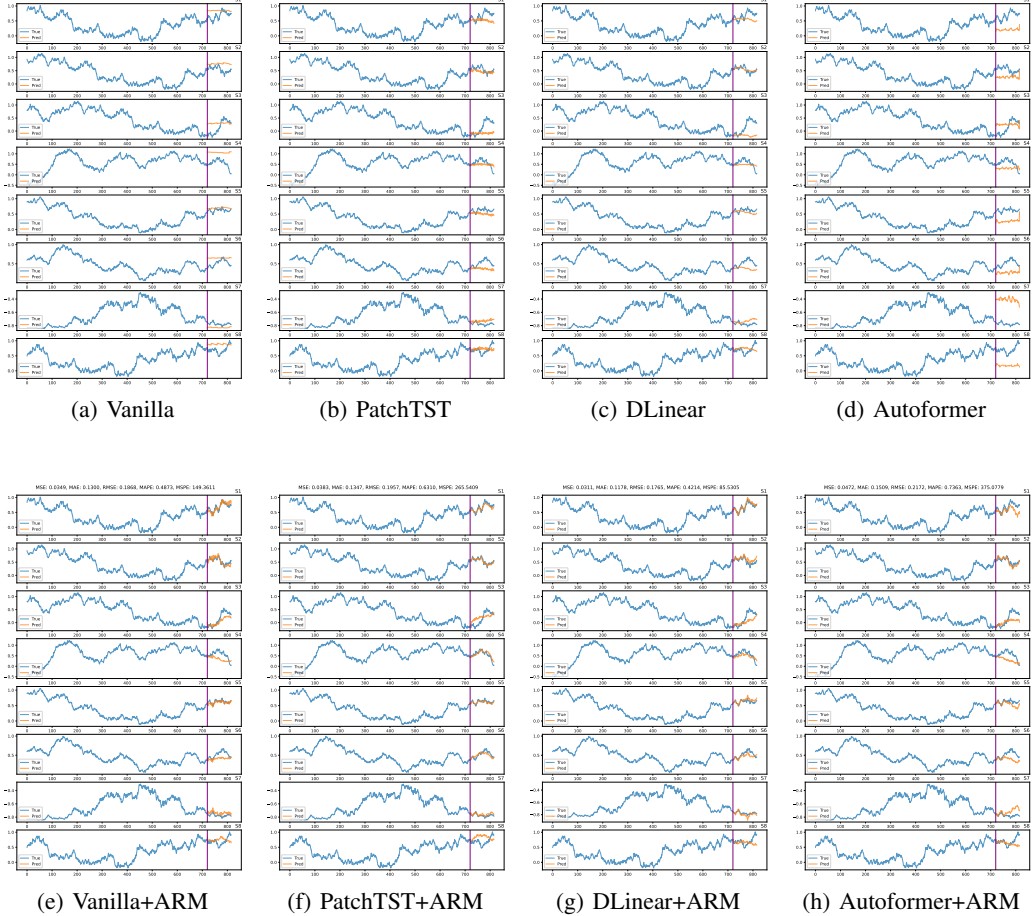

Figure 6: Visualization of the best forecasting results for Vanilla, PatchTST, DLinear, and Autoformer models with and without ARM on the Multi dataset, with a prediction length ($L_P$) of 96 steps. The blue lines show the ground truth time series data and the orange lines denote the 96-step forecasting provided by a specific model. We have randomly selected one sample from the testing set for the visualization of each model. With an input length ($L_I$) of 720 steps, for the 2nd, 3rd, 4th, 5th, 6th, 7th, and 8th time series, we can effectively infer their subsequent 96-step values based on the input of $X^1$. Existing channel-independent univariate models like PatchTST and DLinear fail to model multivariate dependencies, leading to subpar forecasting performance. Furthermore, traditional multivariate LTSF models, like Autoformer, as described earlier, are unable to effectively model the detailed causal relationships between series, leading to similarly unsatisfactory performance. The visualization results show that models with ARM capture the shape of these time series significantly better compared to the models without ARM. Note that the evaluation metrics displayed at the top of each subfigure represent the performance on the entire testing set, not on the current datapoint.

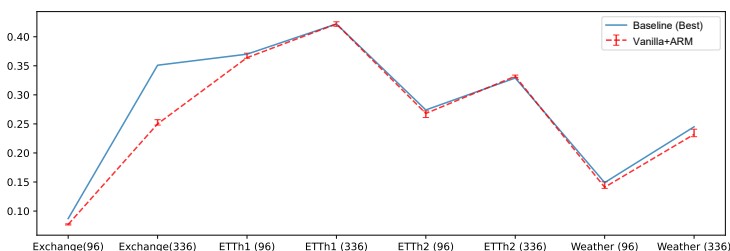

Figure 7: Randomness of Training. The blue line represents the MSE of the combined best baseline results over these eight datasets. The red line, accompanied by corresponding error bars, depicts the best, worst, and average results of ARM (Vanilla) under five settings with random seeds $\in$ $2021, 2022, 2023, 2024, 2025$. On most datasets, the worse results of ARM varying random seeds can still surpass the previously best results from combined baslines.

