# OpenReview forum: "ARM: Refining Multivariate Forecasting with Adaptive Temporal-Contextual Learning"
_ICLR.cc/2024/Conference — ICLR 2024 poster_

### Official Review · Reviewer_X2pk · 2023-10-24

**Soundness:** 3 good
**Presentation:** 3 good
**Contribution:** 4 excellent
**Rating:** 6
**Confidence:** 3

**Summary:**

The paper introduces ARM : a multivariate temporal-contextual adaptive learning method for long-term time series forecasting, which is an enhanced architecture specifically designed for multivariate LTSF modelling. ARM consists 3 modules: Adaptive Univariate Effect Learning (AUEL), Random Dropping (RD) training strategy, and Multi-kernel Local Smoothing (MKLS).  AUEL is for adaptively estimate mean and variance and capture temporal pattern for each series; RD is for robust learning and avoid overfitting when series are interdependent; Multi-kernel Local Smoothing (MKLS) for capturing various temporal dependency among series.  ARM demonstrates superior performance on multiple benchmarks without significantly increasing computational costs compared to vanilla Transformer.

**Strengths:**

Originality: this paper is fairly novel. It attempts to address previous inferior performance of transformer models compared to a simple feed forward neural network DLinear(Zeng et al. 2022) in long term forecasting task. It proposes unique insights on temporal pattern learning of the output sequence distribution and interdependency among univariate series.

Quality: The quality of the paper is high. The proposal of the three core modules are well motivated. AUEL is for adaptively estimate mean and variance and capture temporal pattern for each series; RD is for robust learning and avoid overfitting when series are similar; Multi-kernel Local Smoothing (MKLS) for capturing dependency among series. The authors have performed experiments on 10 benchmarks which are diverse enough to contain different aspect of multivariate time series. The authors also incorporate different ablations by incorporate one or two modules of ARM.

Clarity: the overall clarity of presentation is good. It is motivated by the drawbacks of current sota models and address issues one by one by proposing the modules of ARM.

Significance: the paper can be important to time series learning community.

**Weaknesses:**

I think quality and clarity can be further improved.

Quality: while the empirical results (Table 1 and 2) show superior performance of the proposed model ARM, it will be better to also demonstrate how significant these results are. The authors claim the proposed approach will not significantly increase computational costs. But such discussion/analysis is not included in the paper.

Clarity: I found MKLS block in section 3.3 difficult to follow due to notations. The authors introduced many (subscripts and subscripts of X for example); it would be nice to refer to somewhere.
Table 2 contains large amount of results which are replicate of Table 1. Maybe the authors can eliminate some and find a way to better present them?
Figure 3 & 4. Due to large amount of information in Figures 3&4, when introducing the components such as MKLS block, I think it will be great to introduce pointers to these components in the figures.   For example \tilde{X}^{*i} need pointers to point to in Figure 3.

**Questions:**

1.	How does MoE better capture temporal patterns compared to autoregressive and exponential smoothing models?
2.	What is inverse processing stage?
3.	The paper mentions causal dependency/ relationship among different series. Could the authors explain in what sense the notion of causality is embedded?
4.	I am confused about the how Multi-kernel local smoothing helps capture temporal dependencies among different series due to notations. For example, What is X_j. Could the authors elaborate more?
5.	The paper demonstrates superior empirical performance for forecasting task.  It is hard to see how significant the improvement is compared to other recent models such as PatchTST and DLinear. Do the authors have some insights on the sensitivity of hyperparameters in ARM module in the tuning/training process since many learnable hyperparameters are introduced.
6.	What are the complexity/runtime compute/memory to incorporate these modules in the vanilla model? How does that compare to PatchTST and DLinear?
7.	A few discrepancies regarding Table 1. I found in the original PatchTST paper on traffic datasets Table 3: for MAE,  0.249 for L_P = 96, 0.256 for L_P = 192, different from 0.239 and 0.246 in this paper. Could the authors double check if they are consistent?

---

> ### Author Response · Authors · 2023-11-14
> **Response to Reviewer X2pk (Part 1)**
>
> Thank you very much for your detailed comments on our paper! Your suggestions significantly contribute to enhancing the quality and readability of our research. We appreciate your recognition of the potential value our paper could bring to the community, and we are eager to see our results applied in broader research to create greater value.
>
> Before answering the specific questions, we would like to briefly recap the main ideas of our paper:
>
> - **Overall Concept of ARM**
>
>   i. Utilize AUEL to extract/control each time series' own forecasting contribution.
>
>   ii. Employ Main Predictor + MKLS to model the inter-series relationships beyond univariate forecasting contributions.
>
>   iii. Implement Random Dropping to prevent overfitting when learning inter-series relationships.
>
> ### Q1. About the Presentation Quality of Tables and Figures
> We greatly value your suggestions regarding the presentation of our paper and have made the following adjustments in our revision:
> - We revised the layout of Table 2 to reduce redundancy and added a comparison of average percentage performance improvements with the original model for each sub-experiment to demonstrate the specific enhancements brought by the A/R/M modules.
> - In line with the notations in our paper, we added relevant markings to the Figures of the overall architecture and MKLS block (Figures 2 and 4) to aid in understanding the computational details.
> - We bolded the reference in Section 4 that leads to the discussion on computational costs in Appendix A.7, making it more prominent.
>
> ### Q2. How does MoE Better Capture Temporal Patterns Compared to Autoregressive (AR) and Exponential Smoothing (ES) Models?
>
> - Firstly, the AR and ES methods mentioned in our paper, such as LSTNet and ES-RNN, do not incorporate the AR and ES parameters in the neural network training. In this context, the pre-fixed AR and ES model structures may not suit time series with significant characteristic differences. We added a brief predictability analysis in Appendix A.8 to show the substantial performance variation of these basic learners when dealing with different time series data. In our AUEL, we use an adaptive EMA for each series to adjust the lookback view and then employ MoE to assign the most suitable univariate MLP predictor to each input series, achieving the best adaptation to characteristic differences.
> - Secondly, AR and ES align with the individual univariate model with no parameter sharing described in Figure 3 (a). We consider both this model and the full parameter sharing model in Figure 3 (b) to be suboptimal. If two series in the data have strong relationship (like in the Multi dataset, where one series is a shift of another), parameter sharing between these two series is a good choice. However, if a series in the dataset has no relationship with any other series, sharing its parameters with other series is not a good option. Therefore, we apply the MoE shown in Figure 3 (c) to address this issue, dynamically deciding which predictor from the predictor pool to use for univariate forecasting based on series characteristics.

---

> ### Author Response · Authors · 2023-11-14
> **Response to Reviewer X2pk (Part 2)**
>
> #### Q3. What is the Inverse Processing Stage?
> - In previous methods addressing the input-output distribution shift problem, such as RevIN and NLinear, they estimate each series' output mean and standard deviation based on the input. These values are used to normalize the input, and then applied during the inverse processing stage after the main model processing to ensure that the final output conforms to this distribution. Our AUEL of distribution, positioned between RevIN (full-length lookback) and NLinear (last-step lookback), adaptively adjusts the lookback view for the output distribution estimation, thus following the same computation process above.
> - In the AUEL of temporal patterns, the inverse processing stage helps the model balance between learning intra-series and inter-series dependencies. Note the preprocessing MoE in equation (2) as \\( \\mathtt{MoE}\\left( \\left[ \\widetilde{X}^{\*i}\_I  \\ \\ \\ \\ \\widetilde{X}^{\*i}\_P \\right] \\right) \\) and the inverse processing MoE in equation (3) as \\( \\mathtt{MoE}\\left( \\left[ \\widetilde{X}^{\*i}\_I  \\ \\ \\ \\ \\widehat{X}\_{ED}^{i} \\right] \\right) \\), which only differ in the \\(L\_P\\) forecasting horizon, changing the preprocessed values to the main predictor output \\(\\widehat{X}\_{ED}^{i}\\). If a series \\(i\\) is strongly related to other series and the relationship is effectively captured by the main predictor output, we hope the inverse processing \\( \\mathtt{MoE}\\left( \\left[ \\widetilde{X}^{\*i}\_I  \\ \\ \\ \\ \\widehat{X}\_{ED}^{i} \\right] \\right) \\) maintains this output value or makes minor corrections. However, if a series \\(i\\) has no relation to other series, the inter-series dependencies learned by the main predictor in \\(\\widehat{X}\_{ED}^{i}\\) are likely incorrect. In this case, we expect the inverse processing \\( \\mathtt{MoE}\\left( \\left[ \\widetilde{X}^{\*i}\_I  \\ \\ \\ \\ \\widehat{X}\_{ED}^{i} \\right] \\right) \\) to make significant corrections to the \\(\\widehat{X}\_{ED}^{i}\\) portion, favoring univariate modelling contribution based on \\(\\widetilde{X}^{\*i}\_I\\). On datasets with a large number of series or series from different sources, this correction significantly improves model performance.
>
> #### Q4. About the Causality Mentioned in the Paper
> - Thank you for this valuable question. To avoid any further misunderstanding, we have deleted all the words related to "causality" in the revision. In our previous paper version, we tried to use the term 'causality' to convey the notion that data with strong causal relationships, like those generated through simple shifting operations in the "Multi" dataset, are not effectively modeled by existing models.
> - Indeed, in earlier versions of our paper, we discussed about whether to include a section in the appendix about the connection between ARM and other time series causality studies. You might have noticed, ARM shares some similarities with previous studies like Granger Causality and Vector Autoregressive: firstly control each time series' own forecasting contribution (A) and then explore/test whether other time series provide additional forecasting contributions (Main Predictor + RM). However, since ARM does not perform any statistical tests and differs significantly in practice from these statistical methods, we chose not to include these discussions to avoid misleading.
>
> #### Q5. About the Computational Details in MKLS
> We apologize for the confusion caused. MKLS employs multiple 1D CNNs with different kernel sizes to obtain various local views $\\widetilde{X}_j$ of the input \\(X\\), and uses channel attention to calculate the weight of these local views on each output channel for a weighted average, resulting in a smoothed local view. Therefore, the output of the \\(j\\)th CNN should actually be computed as \\(\\widetilde{X}_j = \\mathtt{Conv1d}\_j \\left(X \\right)\\). Our previous notation of \\(X\\) as \\(X_j\\) was incorrect and caused misunderstanding. This has been corrected in the revision. Please refer to the updated Figure 4 for more details on MKLS computations.

---

> ### Author Response · Authors · 2023-11-14
> **Response to Reviewer X2pk (Part 3)**
>
> ### Q6. Insights on the Sensitivity of Hyperparameters in ARM Module
>
> Thank you for this question. Indeed, the ARM effectively enhances the robustness of LTSF models towards hyperparameter selection. You can refer to section A.5 for our hyperparameter settings. We kept most of the hyperparameters fixed. The only adjustment was that we categorized datasets into small datasets (number of series \\(C < 20\\)) and large datasets (number of series \\(C \geq 20\\)). For small datasets, we used a model dimension of the main predictor \\(d=16\\) and 2 experts in MoE; for large datasets, we used \\(d=64\\) and 4 experts in MoE. Below, we discuss the improvements in hyperparameter selection brought by ARM.
>
> **a. Lookback Length \\(L\_I\\)**
> Previous LTSF models attempted multiple \\(L\_I \\in \\{96, 192, 336, 720\\}\\) during the training phase and chose the optimal one, significantly increasing training costs and making it difficult to select the best \\(L\_I\\) in practical applications. Due to the presence of adaptive EMA, MoE in AUEL, and MKLS, the model can effectively adjust the lookback and local views, making ARM less sensitive to the choice of \\(L_I\\). In all our experiments, we used only \\(L\_I=720\\) and consistently achieved results surpassing previous SOTA models.
>
> **b. Model Dimension \\(d\\) of Main Predictor**
> With ARM, the optimal dimension \\(d\\) required for the main predictor is reduced (b.1), and it can adapt to different datasets with substantial variations in inter-series relationship intensity without adjusting \\(d\\) (b.2).
>
> - **b.1 Reduction of Optimal \\(d\\)**
>    After AUEL controls for univariate forecasting effects, the parameters of the main predictor are primarily used for modeling inter-series dependencies. As mentioned in the response to Q3, MoE distinguishes which series rely mainly on self-forecasting contribution. Therefore, the MKLS-enhanced main predictor only needs to model the remaining inter-series effects, effectively reducing the required \\(d\\). The following table shows the impact of using AUEL of temporal patterns on the optimal \\(d\\) in the Electricity dataset (\\(C=321 > 20\\)) and ETTm1 (\\(C=7 < 20\\)). For detailed analysis, please refer to section A.4.5 and Table 5.
>
>
>
> | Model        | MoE (d=16)      | MoE (d=64)        | MoE (d=256)     | w/o MoE (d=16)   | w/o MoE (d=64)   | w/o MoE (d=256)  |
> |:-:|:-:|:-:|:-:|:-:|:-:|:-:|
> | Metric       | MSE \| MAE      | MSE \| MAE        | MSE \| MAE      | MSE \| MAE      | MSE \| MAE      | MSE \| MAE      |
> | Electricity (96) | 0.128 \| 0.228 | **0.126** \| **0.225** | 0.129 \| 0.229 | 0.313 \| 0.392  | 0.261 \| 0.348  | **0.256** \| **0.341** |
> | Electricity (192) | 0.146 \| 0.242 | **0.142** \| **0.239** | 0.145 \| 0.245 | 0.299 \| 0.378  | 0.255 \| 0.349  | **0.251** \| **0.352** |
> | ETTm1 (96)    | **0.287** \| **0.340** | 0.295 \| 0.348   | 0.300 \| 0.352 | 0.488 \| 0.460  | **0.467** \| **0.446** | 0.503 \| 0.468 |
> | ETTm1 (192)   | **0.325** \| **0.371** | 0.342 \| 0.376   | 0.346 \| 0.385 | 0.531 \| 0.499  | **0.498** \| **0.482** | 0.574 \| 0.512 |

---

> ### Author Response · Authors · 2023-11-14
> **Response to Reviewer X2pk (Part 4)**
>
> **b. Model Dimension \\(d\\) of Main Predictor** (Continue)
>
> - **b.2 Improved Robustness in \\(d\\) Selection**.
> Different datasets, even with the same number of series \\(C\\), can have vastly varying strengths of inter-series relationships. For example, the Exchange and Multi datasets both have \\(C\\) equal to 8, and the series within them exhibit properties similar to random walk processes. However, the former has weak inter-series relationships, while the latter has strong ones.
> After the extraction of univariate effect by AUEL, the model parameters of other parts, whose size mainly controlled by model dimension $d$, will focus on modelling the inter-series dependencies. $d$ typically needs to be tuned to accommodate different inter-series relationship strength of datasets. A smaller $d$ helps avoid overfitting in datasets with weak inter-series relationships, whereas a larger $d$ is necessary for datasets with stronger inter-series dependencies. Incorporating Random Dropping mitigates the need for tuning $d$ across various datasets. Random Dropping can be regarded as an "adaptive scaler" for $d$, allowing the model to automatically adjust to the strength of these relationships. Consequently, $d$ can be set solely based on the number of input series $C$ (we use $d=16$ for $C < 20$ and $d=64$ for $C \geq 20$ for all the experiments), with Random Dropping modulating the learning of inter-series relationships.
> Table below demonstrate the effectiveness of "R": using a **consistent $d=16$** that **potentially fits stronger inter-series dependencies (like Multi)**, models with Random Dropping (**+ARM**) **perform well on dataset with weak inter-series dependencies (Exchange)**, without the **overfitting observed on Exchange** dataset in models without Random Dropping (**+AM**).
>
> | Model            | Autoformer+AM    | Autoformer+ARM  | Informer+AM     | Informer+ARM    | Vanilla+AM      | Vanilla+ARM    |
> |:-:|:-:|:-:|:-:|:-:|:-:|:-:|
> | Metrics          | MSE \| MAE       | MSE \| MAE      | MSE \| MAE      | MSE \| MAE      | MSE \| MAE      | MSE \| MAE     |
> | Exchange (96)    | 0.089 \| 0.210    | 0.081 \| 0.201  | 0.090 \| 0.212   | 0.086 \| 0.207  | 0.085 \| 0.206  | 0.078 \| 0.197 |
> | Exchange (336)   | 0.332 \| 0.417   | 0.298 \| 0.395  | 0.338 \| 0.420   | 0.312 \| 0.404  | 0.304 \| 0.398  | 0.252 \| 0.367 |
> | Average %        | 100% \| 100% | 90.0% \| 95.1%  | 100% \| 100%| 93.0% \| 96.7%  | 100% \| 100%| 84.8% \| 93.4% |
> | Multi (96)       | 0.037 \| 0.130    | 0.038 \| 0.130   | 0.051 \| 0.164  | 0.051 \| 0.166  | 0.034 \| 0.129  | 0.032 \| 0.125 |
> | Multi (336)      | 0.181 \| 0.302   | 0.175 \| 0.300    | 0.182 \| 0.305  | 0.183 \| 0.304  | 0.172 \| 0.291  | 0.164 \| 0.286 |
> | Average %        | 100% \| 100% | 97.7% \| 99.5%  | 100% \| 100%| 100.4% \| 100.2%| 100% \| 100%| 95.1% \| 97.9% |

---

> ### Author Response · Authors · 2023-11-14
> **Response to Reviewer X2pk (Part 5)**
>
> ### Q7. About the Complexity/Runtime Compute/Memory of ARM
>
> We presented a comparison of the computational costs of Vanilla+ARM with previous models in **section A.7** and **Table 7** of the original paper. In the revision, we have added a comparison with DLinear. The results are also presented as follows. It can be observed that under the optimal hyperparameter settings of each model on the ETTm1 dataset, Vanilla+ARM has competetive computational costs compared to previous models.
>
>
> |              | Vanilla+ARM (FLOPs) | Vanilla+ARM (Params) | Vanilla (FLOPs) | Vanilla (Params) | Autoformer (FLOPs) | Autoformer (Params) |
> |:--:|:--:|:--:|:--:|:--:|:--:|:--:|
> | \\(X_P=96\\)   | 426M                | 7.89M                | 244M            | 14.9M            | 10.9G               | 15.5M               |
> | \\(X_P=192\\)  | 515M                | 10.4M                | 273M            | 17.5M            | 11.6G               | 15.5M               |
> | \\(X_P=336\\)  | 664M                | 14.9M                | 316M            | 21.9M            | 12.7G               | 15.5M               |
> | \\(X_P=720\\)  | 1.15G               | 30.6M                | 431M            | 37.8M            | 15.5G               | 15.5M               |
>
> |              | Informer (FLOPs) | Informer (Params) | PatchTST (FLOPs) | PatchTST (Params) | DLinear (FLOPs) | DLinear (Params) |
> |:--:|:--:|:--:|:--:|:--:|:--:|:--:|
> | \\(X_P=96\\)   | 9.41G            | 11.3M             | 5.26G            | 4.87M             | 3.06M           | 138K             |
> | \\(X_P=192\\)  | 10.1G            | 11.3M             | 5.31G            | 7.08M             | 6.10M           | 277K             |
> | \\(X_P=336\\)  | 11.2G            | 11.3M             | 5.38G            | 10.4M             | 10.7M           | 485K             |
> | \\(X_P=720\\)  | 14.0G            | 11.3M             | 5.58G            | 19.2M             | 22.8M           | 1.04M            |
>
>
> ### Q8. Discrepancies in Table 1
>
> We are very grateful for your observation regarding the data entry errors in our paper. We have now corrected these two data points, which also makes our model appear more competitive. We have rechecked the data in Table 1 to ensure its correctness.
>
> ---
> Thank you again for these valuable questions, which greatly contributes to improving the quality of our paper. If you have any further questions, we are eager to engage in more discussions.

---

> > ### Comment · Reviewer_X2pk · 2023-11-20
> > **Thank you for responding.**
> >
> > Thank you for addressing my questions and concerns. I have read through your responses and I would like to increase my score from 5 to 6.

---

### Official Review · Reviewer_Eni6 · 2023-10-26

**Soundness:** 4 excellent
**Presentation:** 2 fair
**Contribution:** 4 excellent
**Rating:** 6
**Confidence:** 4

**Summary:**

The manuscript tackles the multivariate time series forecasting problem, and proposes a solution called ARM.
The proposed method consists of 3 modules, and can be employed to many existing Transformer based time series forecasting models.
Empirical evaluation show that ARM, as well as its 3 modules individually, can improve the forecasting accuracy of the base Transformer model.

**Strengths:**

The idea of applying the 3 modules to existing Transformer models for time series forecasting is novel, to the best of my knowledge.

The text of the manuscript is clear and not difficult to understand.

The proposed model has the potential to contribute to the time series modelling community.

**Weaknesses:**

As mentioned in the Strength part, the text of the manuscript is clear and easy to follow.
However, the figures and tables in the manuscript are very hard to read.
 - Font size of the figures and tables are very small
 - The captions are super long and have a very small font size, I assume this violates the submission guideline.
 - The figures are arranged in an order which is different from how the text refers. Readers have to jump back and forth to find the corresponding (sub-)figure.

**Questions:**

It is not clear to me how MoE is operated from the manuscript.

Why Vanilla+ARM performs better than applying ARM to other Transformer based models?

---

> ### Author Response · Authors · 2023-11-14
> **Response to Reviewer Eni6 (Part 1)**
>
> We are immensely grateful for your recognition and valuable suggestions regarding our paper. Your insights greatly contribute to enhancing the presentation quality of our work. We are thankful for your acknowledgment of the potential value our paper could bring to the community, and we are enthusiastic about the prospect of our findings being applied in broader research to create greater impact.
>
> Before answering the specific questions, we would like to briefly recap the main ideas of our paper:
>
> - **Overall Concept of ARM**
>
>   i. Utilize AUEL to extract/control each time series' own forecasting contribution.
>
>   ii. Employ Main Predictor + MKLS to model the inter-series dependencies beyond univariate forecasting contributions.
>
>   iii. Implement Random Dropping to prevent overfitting when learning inter-series dependencies.
>
>
> ### Q1. About the Formatting Problems in the Paper
> We sincerely appreciate your suggestions regarding the formatting of our paper. Following your advice, we have made modifications to significantly enhance its readability.
>
> a. We have increased the font size of Tables 1 and 2 and adjusted the layout of Table 2 to make it more readable.
>
> b. All captions have been resized to a standard font size and have been succinctly revised for clarity.
>
> c. The positions of Figures 2 and 3 have been swapped for a more appropriate presentation.

---

> ### Author Response · Authors · 2023-11-14
> **Response to Reviewer Eni6 (Part 2)**
>
> ### Q2. About the Operation Details of MoE
>
> In AUEL, the MoE is responsible for building univariate temporal patterns. It adaptively allocates the most suitable univariate MLP predictor for each series. Here are the benefits of using MoE:
> - Between the individual univariate models in Figure 3 (a) and the fully-shared-parameter models in Figure 3 (b), MoE in Figure 3 (c) can adapt to datasets with both similar and independent series.
> - The presence of MoE allows the main predictor to focus solely on the contributions of inter-series dependencies, thereby reducing the required number of parameters for the main predictor.
> - In inverse processing, MoE can modify the output of the main predictor, choosing whether to retain the forecasting with inter-series information modeled by the main predictor or to correct it to rely solely on the univariate forecasting of the current series.
>
> We illustrate the input and output of MoE in Equations (2) and (3). In the preprocessing stage, for each series $i$, MoE builds temporal patterns for the $L_P$ part containing only constant values, based on the processed $\\widetilde{X}^{\*i}$ from the AUEL of distribution. This preprocessing stage of AUEL of temporal patterns is computed as $ \\widetilde{X}^{i} = \\left\[ \\widetilde{X}^{\*i}\_I  \\ \\ \\ \\ \\mathtt{MoE}\\left( \\left[ \\widetilde{X}^{\*i}\_I  \\ \\ \\ \\ \\widetilde{X}^{\*i}\_P \\right] \\right) \\right\] $. In the inverse processing stage, MoE determines whether to modify the output based on the input $ \\widetilde{X}^{i}\_I = \\widetilde{X}^{\*i}\_I $ of the main predictor and its output $ \\widehat{X}\_{ED}^{i} $, to balance learning intra-series and inter-series dependencies. This inverse processing stage of AUEL of temporal patterns is calculated as $\\widehat{X}\_M^{i} = \\mathtt{MoE}\\left( \\left[ \\widetilde{X}^{\*i}\_I  \\ \\ \\ \\ \\widehat{X}\_{ED}^{i} \\right] \\right)$.
>
> For the specific computations within the MoE block, we employed the model structure from the Switch Transformer (Fedus et al., 2022), which uses routing operations to match each input series to the corresponding MLP predictor for computation. We also used residual connections on $L_P$ to ensure stable training. Details about these computation methods are described in section **A.6.1** of our paper, which we have also emphasized in the revision by bolding the reference in section 3.1. The description is presented as follows:
>
> > For adaptive learning of temporal patterns, we utilize the MoE predictor structure as described in Switch Transformer (Fedus et al., 2022). We adjusted the output dimension of its final layer to cater to the input length of $L_I+L_P$, an output length of $L_P$, with a hidden dimension of $4(L_I+L_P)$. Given the discrepancy in input and output lengths, the MoE's residual shortcut needs specific modifications. We establish a residual connection using the \\(L_P\\) section of the input before the AUEL preprocessing and before the AUEL inverse processing. Specifically, in the calculation of MoE in AUEL preprocessing, where $\\widetilde{X}^{i} = \\left[ \\widetilde{X}^{\*i}\_I \\ \\ \\mathtt{MoE}\\left( \\left[ \\widetilde{X}^{*i}\_I \\ \\ \\widetilde{X}^{\*i}\_P \\right] \\right) \\right]$, we actually compute \\(\\mathtt{MoE}\\left( \\left[ \\widetilde{X}^{\*i}\_I \\ \\ \\widetilde{X}^{\*i}\_P \\right] \\right)=\\mathtt{MLP}\_{\\mathtt{selected}}\\left( \\left[ \\widetilde{X}^{\*i}\_I \\ \\ \\widetilde{X}^{\*i}\_P \\right] \\right) + \\mathbf{0}^i\_P\\). Here, \\(\\mathtt{MLP}\_{\\mathtt{selected}}\\) denotes the MLP predictor selected by the routing part of MoE based on the input, and \\(\\mathbf{0}^i\_P\\) is the \\(L\_P\\) part before AUEL preprocessing, i.e., the default 0 values before AUEL. During inverse processing, we compute \\(\\widehat{X}\_M^{i}=\\mathtt{MoE}\\left( \\left[ \\widetilde{X}^{\*i}\_I \\ \\ \\widehat{X}\_{ED}^{i} \\right] \\right)=\\mathtt{MLP}\_{\\mathtt{selected}}\\left( \\left[ \\widetilde{X}^{\*i}\_I \\ \\ \\widehat{X}\_{ED}^{i} \\right] \\right)+\\widehat{X}\_{ED}^{i}\\), where the added \\(\\widehat{X}\_{ED}^{i}\\) is the \\(L_P\\) part before the AUEL block.

---

> ### Author Response · Authors · 2023-11-14
> **Response to Reviewer Eni6 (Part 3)**
>
> ### Q3. Why Vanilla+ARM Performs Better than Applying ARM to Other Transformer-Based Models
>
> Thank you for your insightful question. Indeed, this is a scenario we anticipated during our model design process. Once ARM is applied, the primary role of the main predictor shifts to modeling inter-series dependencies. With the assistance of MKLS, a structurally simple yet effective predictor is required to process the reasonable multivariate representation constructed by MKLS. In cases where token representation is well-designed, the vanilla Transformer has been proven adaptable to a wide range of tasks. Autoformer, on the other hand, introduces an auto-correlation mechanism in the temporal dimension, an improvement that does not align with our goal of modeling inter-series relationships. Informer introduces sparse attention to increase computational efficiency, but this actually compromises performance compared to full attention. Nevertheless, it is worthy to note that, as seen in Table 2, Autoformer+ARM and Informer+ARM still demonstrate stable improvements over past SOTA models like PatchTST and DLinear without ARM.
>
> ---
> We thank you again for your valuable question, which greatly contributes to improving the quality of our paper. If you have any further questions, we are keen to engage in more discussions.

---

> ### Comment · Reviewer_Eni6 · 2023-11-14
>
> Thanks to the authors for updating the manuscript and for answering my questions. I will keep my positive rating.
>
> Some of the references from arXiv have been published already, e.g. Zhanghao Wu et al. 2020 is published at ICLR, and Aurko Roy (don't know what the star means) et al. 2020 at *Transactions of the Association for Computational Linguistics*.

---

> > ### Author Response · Authors · 2023-11-22
> >
> > Dear Reviewer Eni6,
> >
> > Thank you so much for your constructive comments and for maintaining a positive rating on our research. We have carefully reviewed and updated all references in the revised paper to ensure they reflect the published versions. We appreciate your attention to detail. Thank you!

---

### Official Review · Reviewer_PUmd · 2023-11-01

**Soundness:** 3 good
**Presentation:** 2 fair
**Contribution:** 3 good
**Rating:** 6
**Confidence:** 3

**Summary:**

The paper proposes an enhanced architecture for multivariate time series forecasting using Transformers. The proposed ARM approach incorporates three innovations - Adaptive Univariate Effect Learning (AUEL), Random Dropping (RD), and Multi-kernel Local Smoothing (MKLS). AUEL component introduces learnable exponential moving average instead of classic autoregressive and exponential smoothing for initiating prediction part into the encoder. Random Dropping is almost like an ensemble model which models selected subsets of time series, with aim to reduce spurious patterns among the timeseries. And MKLS which uses one-dimensional convolutional kernels and a channel-wise attention to encapsulate local information. The approach is evaluated on multiple datasets against several competitor approaches.

**Strengths:**

Consistently overperforms multiple strong competitor approaches on several datasets.
Ablation study is assessing effects of each of the three presented components.

**Weaknesses:**

Some of the performances in the Table one are reported from their respective papers, so I am wondering if it is likely to assure the same experimental setup.
Certain statements are presented without appropriate evidence to support the claim. For example 'we introduce ARM, a methodology designed for correctly training multivariate LTSF models.', which is strong claim, and moreover implies that other approaches are 'incorrectly training'.

**Questions:**

N/A

---

> ### Author Response · Authors · 2023-11-14
>
> We greatly appreciate your questions and comments on our paper. Your insights are invaluable to us.
>
> Before answering the specific questions, we would like to briefly recap the main ideas of our paper:
>
> - **Overall Concept of ARM**
>
>   i. Utilize AUEL to extract/control each time series' own forecasting contribution.
>
>   ii. Employ Main Predictor + MKLS to model the inter-series dependencies beyond univariate forecasting contributions.
>
>   iii. Implement Random Dropping to prevent overfitting when learning inter-series dependencies.
>
> ### Q1. About the Consistency of Experimental Setup
> The experimental setup for previous baseline models, including PatchTST, DLinear, FedFormer, Autoformer [1-4], shared the same data processing methods and experimental environment. We built our model code on this same codebase to maintain consistency in the environment. Therefore, in Table 1, we could opt to quote previously reported results from SOTA models, avoiding the potential random errors and fairness concerns that might arise from rerunning experiments. It’s important to note that previous SOTA models reported optimal results for experimental settings with input length $L_I \in \\{ 96, 192, 336, 720 \\} $. However, models with our ARM is less sensitive to the choice of \\( L_I \\), allowing us to consistently use \\( L_I=720 \\) for all experiments. Even under these comparatively challenging conditions, our results still surpassed previous SOTAs.
>
> Conversely, for Table 2, we reran all baseline models at \\( L_I=720 \\) to demonstrate the improvements brought by integrating ARM. We imported their model structures directly from the baseline models' official codes listed below (see `ARM.py` in supplementary materials) and reran these experiments using hyperparameter settings obtained from their provided scripts. The effective performance improvements from ARM under the condition of merely altering the addition of A/R/M modules provide a fair comparison of results through this ablation method.
>
> In summary, in both quoted version (Table 1) and reproduced version (Table 2), ARM stably surpuss the previous SOTA results.
>
> [1] https://github.com/yuqinie98/PatchTST/tree/main/PatchTST_supervised/data_provider
>
> [2] https://github.com/cure-lab/LTSF-Linear/tree/main/data_provider
>
> [3] https://github.com/MAZiqing/FEDformer/tree/master/data_provider
>
> [4] https://github.com/thuml/Autoformer/tree/main/data_provider
>
>
> ### Q2. About the Inappropriate Strong Claim
> Thank you for your attention to the rigor of our paper. In the revision, we have changed "correctly" to "effectively" in response to your concern. It's worth noting that our initial use of "correctly" was based on our considerations. As illustrated in the paper, our idea builds upon the analysis of past observations:
> - Univariate models like PatchTST and DLinear significantly outperformed previous multivariate LTSF models, possibly indicating:
>     - (1) Previous multivariate models failed to properly handle multivariate time series input, resulting in inferior performance in modeling intra-series temporal dependencies compared to univariate models.
>     - (2) Weak or hard-to-model inter-series dependencies in existing benchmark datasets, possibly leading to better performance of univariate models, while multivariate models might learn wrong patterns or tend to overfit.
> - Based on these analyses, we designed ARM to help LTSF models address these issues. Furthermore, we synthesized a Multi dataset with clear inter-series dependencies, generated based on simple shifting of series, as an example. This illustrates that even in scenarios with obvious dependencies, previous multivariate models still failed to learn basic "copy-paste" operations (see Figure 1, section A.2, and Figure 6). Since existing LTSF models struggle with such simple series-wise dependency learning, we believe they "incorrectly process the multivariate time series input".
>
> Considering these analyses was based on these observations and to avoid potential misunderstandings and overgeneralizations, we decided to replace "correctly" with "effectively" in the revision.
>
> ---
> Thank you again for your questions and suggestions! You might have additional queries about our paper, and we are more than happy to discuss and clarify them. Your feedback helps a lot in enhancing the quality of our research.

---

> > ### Comment · Reviewer_PUmd · 2023-11-20
> >
> > Thanks to authors for detailed responses, after reading the overall comments and feedback, I have increased the rating from 5 to 6.

---

### Author Response · Authors · 2023-11-14
**Summary**

Dear reviewers,

We sincerely thank you for your insightful comments and valuable suggestions, which have significantly enhanced the quality of our research.

We are grateful that all three reviewers recognized that our model's performance consistently surpasses previous SOTA. We appreciate Reviewer Eni6 and X2pk's acknowledgment of the quality of our paper and their belief in its potential contribution to the community. Given that the A/R/M modules can be easily transferred to other models with stable performance improvements, we hope our findings can be applied more broadly in both research and practice. We have already provided the current model structure code of ARM in the supplementary materials and will share the complete codebase once the paper decision is made.

The overall concept of ARM includes: i) Utilizing AUEL to extract/control each time series' own forecasting contribution; ii) Employing Main Predictor + MKLS to model the inter-series dependencies beyond univariate forecasting contributions; iii) Implementing Random Dropping to prevent overfitting when learning inter-series dependencies. ARM empowers LTSF models to better handle multivariate time series input with characteristic differences, thereby enabling them to effectively model inter-series dependencies, which was a challenge for previous LTSF models.

In response to the reviewers' suggestions on our paper's formatting, presentation, and readability, we have made the following key updates:

1. Paper Format and Correction
- We modified the overall size, font size, and captions of Figures and Tables to improve readability. We simplified some of the redundant expressions in the paper.
- We added average percentage comparisons in Table 2 to show specific improvements each baseline model gained by adding A/R/M.
- We rearranged some Figures to follow the reading sequence of the paper. We corrected some notations and placed some of the notations in Figures for easier understanding.
- We highlighted some reference links to guide readers to section A.6 for additional implementation details and section A.7 for computational costs.
- We revised some phrases to avoid misleading readers. We rechecked the data in the results section to ensure its accuracy.

2. Additional Insights and Expanded Analysis
- We expanded the analysis in A.4.5 on optimal model size reduction and robustness improvement of model dimension selection brought by ARM, supported by additional experiments.
- In A.7, we added more baselines for computational cost comparisons.
- In A.8, we included a predictability analysis of the datasets we studied, showing substantial performance differences when using the same basic time series learners on different datasets with varying series characteristics, highlighting the importance of adaptive design in ARM.

We again thank the reviewers for their valuable suggestions on our paper. If you have more questions about the paper, we are very eager to discuss them further and clarify any doubts you may have. We are thankful for the opportunity to refine our paper and are open to any further queries or discussions that may enhance its contribution to the field.

---

### Meta-Review · Area_Chair_xUD2 · 2023-12-05

**Metareview:**

The paper proposes ARM, a novel approach for multivariate long-term time series forecasting using Transformers. It incorporates Adaptive Univariate Effect Learning, Random Dropping, and Multi-kernel Local Smoothing. These modules have been comprehensively analyzed and demonstrated to yield strong forecasting performance. The design ideas behind these can be potentially impactful to the time series forecasting community.

Overall, the strengths indicated below overweigh the concerns, and I suggest the acceptance of the paper. Please improve the submission with the answers provided to reviewer questions, especially on the points indicated as weaknesses.

**Justification For Why Not Higher Score:**

- Certain claims lack sufficient evidence.
- Concerns with benchmarking setup
- Insufficient analysis on computational cost.
- Figure quality

**Justification For Why Not Lower Score:**

- Consistently outperforms strong competitors on several diverse benchmarks.
- Motivated by clear limitations of existing models and addresses them with distinct modules.
- Ablation study assesses the impact of each module.
- Addresses the poor performance of Transformers in long-term forecasting compared to simpler models. Novel idea of applying the modules to existing Transformer models.
- Clear and easy-to-understand text.
- Potential for significant impact on the time series learning community.

---

### Decision · Program_Chairs · 2024-01-16

Accept (poster)